# Evaluating Bivariate Causal Statements Based on Mutual Compatibility

**Erik Jahn** [1 2]  **Dominik Janzing** [2]

## Abstract

For many real-world systems, causal ground truth is difficult to obtain, making claims about causal effects hard to assess. We develop methods for evaluating collections of bivariate causal statements, one for each pair of variables in a fixed system. In the setting of acyclic linear statements, any such collection can be extended to a unique multivariate causal model, but we argue that this induced model is implausible if it imposes substantial additional confounding to explain observed correlations. We introduce a compatibility score that quantifies this notion of plausibility, notably without relying on the faithfulness assumption. Additionally, we define an incompatibility score for purely graphical bivariate causal statements, based on global consistency constraints that are derived from acyclicity and faithfulness assumptions. We give theoretical and empirical evidence that both scores can successfully distinguish correct from incorrect causal statements in generic settings. Moreover, we demonstrate the practical applicability of our methods by analyzing causal claims made by large language models. Our work aims to provide a foundation for assessing the reliability of causal information derived from human experts or artificial intelligence in settings where alternative forms of validation are unavailable.

## 1. Introduction

Causal inference in many domains such as medicine or economics still relies heavily on expert knowledge (Didelez, 2024; Gururaghavendran & Murray, 2024; Atanasov & Black, 2016). Such knowledge is often expressed in the form of pairwise cause-effect statements. Indeed, there is empirical evidence that humans find it easier to assess

[1]California Institute of Technology, USA [2]Amazon Research Tübingen, Germany. Correspondence to: Erik Jahn <ejahn@caltech.edu>.

*Proceedings of the 43$^{rd}$ International Conference on Machine Learning*, Seoul, South Korea. PMLR 306, 2026. Copyright 2026 by the author(s).

bivariate causal relations than to infer entire multivariate causal graphs (Tatlidil et al., 2025). A similar pattern can be observed when querying large language models (LLMs) for causal relationships (Kiciman et al., 2024; Sheth et al., 2025), an approach that has attracted substantial recent attention (Ma, 2025; Yu et al., 2025; Wan et al., 2025). While LLMs have shown promising performance on some causal reasoning tasks, there are virtually no guarantees regarding the quality of their outputs. Unlike most causal discovery algorithms, which at least come with well-understood consistency guarantees under strong assumptions, most LLMs are not specifically adapted or trained for accurate causal inference (Zečević et al., 2023). Causal statements from humans can be subject to a variety of well-known biases and are not always based on objective data (Tversky & Kahneman, 1974). This raises a fundamental question: how can we ever trust human- or AI-generated causal statements?

In many real-world settings, obtaining causal ground truth through experimentation is either infeasible or prohibitively expensive, leaving no straightforward answer. The few existing approaches for evaluating causal statements in the absence of ground truth are mostly based on defining and measuring different notions of consistency between an estimated causal graph and the data (Textor et al., 2016; Eulig et al., 2025; Sheth et al., 2026). In contrast, Faller et al. (2024) consider the *compatibility* of multiple estimated causal graphs across different subsets of variables to evaluate causal graph learning algorithms. In this paper, we focus on the setting, where only bivariate causal statements are available, and develop notions of compatibility for evaluating and falsifying such statements.

We pursue this approach at two levels of causal information. Given the prevalence of linearity assumptions in causal inference (Zanga et al., 2022), we first focus on falsifying bivariate causal statements that quantify total pairwise causal effects by a linear causal coefficient. In the absence of faithfulness, we show that for any acyclic list of such statements, there exists a unique multivariate causal model that perfectly fits the data and whose marginalizations exactly coincide with the given statements. As a consequence, there are no hard compatibility constraints in the sense of previous work for such lists, beyond consistency with an acyclic causal ordering. We therefore propose to measure compatibility based on the *plausibility* of the unique induced multivariate

causal model. We postulate that a plausible multivariate causal model should exhibit less confounding than its bivariate marginal models, as correlations induced by observed causal backdoor paths in the multivariate model become unobserved confounding in the marginals. We formalize this intuition mathematically, leading to a measure of model plausibility. Theoretically and empirically, we demonstrate that this measure successfully distinguishes correct from incorrect bivariate causal statements. We further apply our approach to causal statements obtained from LLMs for socioeconomic indicators and show that the resulting scores vary substantially across different models and strongly correlate with general model performance.

Second, we consider the case where only qualitative bivariate causal statements are available, indicating the presence or absence of a causal effect and of confounding. Under the faithfulness assumption, such statements are subject to hard graphical compatibility constraints, following Faller et al. (2024). In this setting, we develop an incompatibility score that measures the number of violations of these constraints. In synthetic experiments we investigate how the score reflects the true number of errors in a list of bivariate causal statements and demonstrate our approach for statements generated by LLMs.

# 2. Compatibility for Linear Bivariate Causal Statements

## 2.1. Linear Structural Equation Models

In this section, the causal models we consider are *linear structural equation models (SEMs)*. A linear SEM for a vector of observable random variables $\mathbf{X} = (X_1, \ldots X_n)$ is specified by a noise vector of random variables $\mathbf{N} = (N_1, \ldots, N_n)$ and a matrix of causal coefficients $\Gamma \in \mathbb{R}^{n \times n}$. We assume *acyclicity* and require $\Gamma$ to be strictly lower-triangular. Then, the model is given by

$$\mathbf{X} = \Gamma \mathbf{X} + \mathbf{N}. \tag{1}$$

Since $\Gamma$ is lower-triangular, $(I - \Gamma)$ is invertible, so $\mathbf{X} = (I - \Gamma)^{-1} \mathbf{N}$ is the unique solution to equation (1). We do not assume the entries of the noise vector to be independent, therefore allowing for correlations representing unmeasured confounding.

The causal interpretation of SEMs is that they define the distribution of $\mathbf{X}$ under arbitrary *interventions* (or experiments). An intervention sets variables $X_j$ for $j \in S \subseteq \{1, \ldots, n\}$ to fixed values $x_j \in \mathbb{R}$. Then, the remaining variables can be found from solving the equation

$$\mathbf{X} = I_{\overline{S}} \Gamma \mathbf{X} + I_{\overline{S}} \mathbf{N} + \mathbf{x}_S. \tag{2}$$

Here, $I_{\overline{S}}$ is the diagonal matrix that has 1's on the diagonal for indices in the complement $\overline{S}$ of $S$, and zeros otherwise,

and $\mathbf{x}_S$ has entries $x_j$ for $j \in S$ and zeros elsewhere. We denote the resulting interventional probability distribution of the variables solving equation (2) by $P(\mathbf{X} \mid \mathrm{do} \, \mathbf{X}_S = \mathbf{x}_S)$. Linear SEMs are closed under marginalization in the following sense:

**Lemma 2.1** (marginalized SEM). *Let* $\mathbf{X} = \Gamma \mathbf{X} + \mathbf{N}$ *be a linear SEM and let* $\mathbf{Y}, \mathbf{Z}$ *be a partition of* $\mathbf{X}$. *Then, there exists a noise vector* $\tilde{\mathbf{N}}$ *such that the SEM*

$$\mathbf{Y} = \left( \Gamma_{\mathbf{YY}} + \Gamma_{\mathbf{YZ}} (I - \Gamma_{\mathbf{ZZ}})^{-1} \Gamma_{\mathbf{ZY}} \right) \mathbf{Y} + \tilde{\mathbf{N}}.$$

*induces the same observational and interventional distributions as the original model, marginalized to* $\mathbf{Y}$.

Here, $\Gamma_{\mathbf{YZ}}$ denotes the submatrix of $\Gamma$ whose rows match the indices of variables in $\mathbf{Y}$ and whose columns match the indices of variables in $\mathbf{Z}$. For a proof of Lemma 2.1, see for instance Lemma 7 of Hyttinen et al. (2012).

## 2.2. Linear Bivariate Causal Statements

**Definition 2.2** (Bivariate causal statements). A *linear bivariate causal statement* for a pair $(X_i, X_j)$ specifies

1. a causal direction, e.g. $i \to j$;

2. a linear causal coefficient $\alpha_{ij} \in \mathbb{R}$.

Our goal is to evaluate a complete list of $\binom{n}{2}$ linear bivariate causal statements for a fixed set of variables $\{X_1, \ldots, X_n\}$ based on their mutual compatibility. After potentially renaming variables, we assume that all proposed causal directions are compatible with the ordering of $X_1, \ldots, X_n$ given by their indices, that is, each statement with a nonzero coefficient says $i \to j$ for $i < j$. Note that this ordering need not be a correct causal ordering. We further assume that the joint probability distribution of $(X_1, \ldots, X_n)$ is known. Then, a linear bivariate causal statement assigning the causal coefficient $\alpha_{ij}$ to $(X_i, X_j)$ equivalently proposes the linear bivariate SEM

$$X_j = \alpha_{ij} X_i + \tilde{N}_{ij}, \tag{3}$$

where the distribution of $\tilde{N}_{ij}$ is just defined as the distribution of $X_j - \alpha_{ij} X_i$. Crucially, when all proposed causal directions respect a common ordering, there are no further *hard compatibility constraints*: any complete list of linear bivariate causal statements induces a unique multivariate linear SEM whose pairwise marginals agree with the proposed bivariate SEMs. We encode a list of $\binom{n}{2}$ bivariate statements by the unit lower-triangular matrix $A \in \mathbb{R}^{n \times n}$ with entries

$$A_{ii} = 1, \qquad A_{ij} = \alpha_{ji} \; (i > j), \qquad A_{ij} = 0 \; (i < j).$$

**Lemma 2.3** (Existence of a unique compatible SEM). *Let $A$ be a unit lower-triangular matrix of linear bivariate causal statements for a vector $\mathbf{X}$ of observed variables and define $\Gamma = I - A^{-1}$. Then, the multivariate structural equation model*

$$\mathbf{X} = \Gamma\mathbf{X} + \mathbf{N},$$

*where the distribution of $\mathbf{N}$ is defined as the distribution of $(I - \Gamma)\mathbf{X}$, is the unique linear SEM whose pairwise marginal submodels are*

$$X_j = A_{ji}X_i + \tilde{N}_{ij}.$$

The proof of Lemma 2.3 only uses basic linear algebra and we provide it in Appendix B.

### 2.3. Confounding Postulate

Given that any acyclic collection of linear bivariate causal statements is compatible with a unique multivariate SEM, we propose to evaluate such collections based on the *plausibility* of the induced multivariate model. Clearly, there is no unique way to define plausibility. In this work, we suggest a necessary criterion for plausibility based on the following postulate:

**Assumption 2.4** (Confounding postulate). *A generic multivariate causal model should not exhibit more confounding than its pairwise marginal models.*

Here, confounding corresponds to correlation between variables that cannot be explained by observed causal effects, and we will formalize this notion in the next section. Intuitively, marginalizing a multivariate causal model to a pairwise model drastically decreases the number of observed causal effects and therefore the amount of confounding in the marginal model should increase. This intuition only fails when the multivariate model is specifically designed so that the correlation induced by its additional observed causal pathways cancels with the multivariate confounding (see Figure 1). Ruling out such cancellations is reminiscent of the standard faithfulness condition (see Appendix A), but our postulate is logically independent from it. While some near-cancellations of causal pathways can randomly appear (Uhler et al., 2013), a lot more fine-tuning is required for the global confounding of a large multivariate model to be smaller than that of all its pairwise marginals. We provide a theoretical foundation for this claim in Section 2.6 by showing that random causal models satisfy Assumption 2.4 in expectation, and in Section 2.7, we demonstrate empirically that this is also true with high probability.

### 2.4. Confounding Measures

To formalize Assumption 2.4, we need to quantify the amount of confounding in a causal model. There is no universally accepted confounding measure (see Reddy & Balasubramanian (2024) for some options). In principle, our approach of falsifying linear bivariate causal statements based on Assumption 2.4 can be implemented with different choices of confounding measures. We simply need one that enables a fair comparison between a multivariate model and its pairwise marginals. It turns out that quantifying confounding based on squared parts of pairwise covariances between variables that are not explained by observable causal pathways works well for our purpose. Let $\Sigma$ denote the covariance matrix of a vector of observable variables $\mathbf{X}$. In a linear bivariate SEM $X_j = \alpha_{ij}X_i + \tilde{N}_{ij}$, the covariance decomposes as

$$\Sigma_{ij} = \alpha_{ij}\Sigma_{ii} + \text{Cov}(X_i, \tilde{N}_{ij}).$$

The first term describes the covariance due to the causal effect from $X_i$ to $X_j$, and the second is due to confounding. This motivates the following definition:

**Definition 2.5** (amount of confounding in bivariate models). Let $\Sigma$ be a covariance matrix and let $\alpha_{ij}$ be a proposed bivariate causal coefficient from $X_i$ to $X_j$. We define the bivariate amount of confounding for the pair $(i, j)$ as

$$\mathcal{C}_{ij}^{\text{biv}}(\Sigma, \alpha_{ij}) = (\Sigma_{ij} - \alpha_{ij}\Sigma_{ii})^2.$$

Here, we use squaring to get a confounding score which is non-negative, and equals zero if and only if the noise term for $X_j$ is uncorrelated with $X_i$. Up to normalization, this score has been studied before (Janzing & Schölkopf, 2018a;b), but only in the bivariate setting. We now extend this idea to multivariate models. Consider a linear SEM $\mathbf{X} = \Gamma\mathbf{X} + \mathbf{N}$ on $n$ variables with covariance matrix $\Sigma = \text{Cov}(\mathbf{X})$. For an ordered tuple $P = (t_1, \ldots, t_m)$ with $1 \le t_0 < \cdots < t_m \le n$, we define

$$\Gamma^P = \prod_{s=0}^{m-1} \Gamma_{t_{s+1}, t_s}$$

with the convention that the empty tuple $P_0 = ()$ gives $\Gamma^{P_0} = 1$. The tuple $P$ can be thought of as specifying a directed causal path $X_{t_0} \to \cdots \to X_{t_m}$ and $\Gamma^P$ is the product of causal coefficients along the path. By Wright's path tracing rules (Wright, 1934), the covariance between $X_i$ and $X_j$ decomposes as

$$\Sigma_{ij} = \sum_{k \le i} \Sigma_{kk} \cdot \sum_{\substack{P_1:k \rightsquigarrow i, P_2:k \rightsquigarrow j, \\ P_1 \cap P_2 = \emptyset}} \Gamma^{P_1}\Gamma^{P_2}$$

$$+ \sum_{\ell \ne k \le j} \text{Cov}(N_\ell, N_k) \cdot \sum_{\substack{P_1:\ell \rightsquigarrow i, P_2:k \rightsquigarrow j \\ P_1 \cap P_2 = \emptyset}} \Gamma^{P_1}\Gamma^{P_2} \quad (4)$$

Here, each $P : k \rightsquigarrow i$ runs over all directed paths from $k$ to $i$, and two paths are disjoint if they share no vertices,

Covariance matrix

$$\Sigma_{\mathbf{X}} = \begin{pmatrix} 1 & 0.5 & 0.5 \\ 0.5 & 1 & 0.5 \\ 0.5 & 0.5 & 1 \end{pmatrix}$$

Bivariate marginal SEMs

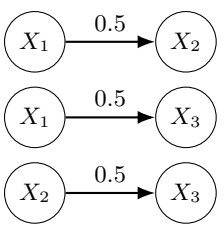

Multivariate SEM

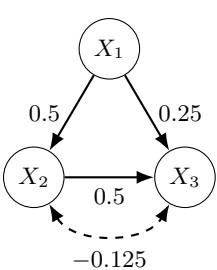

*Figure 1.* Example of linear bivariate causal statements inducing a trivariate model that violates Assumption 2.4. Here, the bivariate causal effects exactly account for the pairwise covariances, which results in all marginal models having confounding scores 0. However, the induced trivariate SEM exhibits confounding between $X_2$ and $X_3$ that precisely cancels with the back-door path $X_2 \leftarrow X_1 \rightarrow X_3$ in order to match both the covariance matrix and the marginal causal coefficients. Hence, the trivariate model has a confounding score of $(-0.125)^2$, resulting in a negative compatibility score. Intuitively, it should be implausible to proclaim strong causal effects between all three variables, but ignoring the fact that the back-door path $X_2 \leftarrow X_1 \rightarrow X_3$ then induces confounding in the marginal SEM on $(X_2, X_3)$.

except possibly endpoints. The first term in equation (4) quantifies the contribution of all direct causal paths and fully observed back-door paths to the covariance of $X_i$ and $X_j$, whereas the second term is the part of the covariance that cannot be explained by observed causal effects but is due to correlation of the noise variables.

**Definition 2.6** (amount of confounding in multivariate models). Let $\Sigma$ be a covariance matrix and let $\Gamma$ be the lower-triangular causal coefficient matrix of a multivariate linear SEM. For $i < j$, we define the multivariate amount of confounding for the pair $(i, j)$ as

$$\mathcal{C}_{ij}^{\text{mult}}(\Sigma, \Gamma) = \left( \Sigma_{ij} - \sum_{k \leq i} \Sigma_{kk} \sum_{\substack{P_1 : k \rightsquigarrow i, P_2 : k \rightsquigarrow j, \\ P_1 \cap P_2 = \emptyset}} \Gamma^{P_1} \Gamma^{P_2} \right)^2.$$

For two variables, this definition coincides exactly with Definition 2.5.

### 2.5. Compatibility and Falsification

We can now quantify the compatibility of a list of linear bivariate causal statements given by a matrix $A$ as the degree to which Assumption 2.4 holds for the unique induced multivariate model with causal coefficient matrix $\Gamma = I - A^{-1}$ (see Lemma 2.3).

**Definition 2.7** (compatibility score). For a list of $n$ observable variables, let $\Sigma$ be the covariance matrix and $A$ be a unit lower-triangular matrix of linear bivariate causal statements. We define the *compatibility score*:

$$\text{comp}(\Sigma, A) = \sum_{1 \leq i < j \leq n} \mathcal{C}_{ij}^{\text{biv}}(\Sigma, A_{ji}) - \mathcal{C}_{ij}^{\text{mult}}(\Sigma, I - A^{-1}).$$

The exact size of the compatibility score depends on the scaling of the variables. Therefore, in practice, we compute the score after standardizing all variables to unit variance, equivalently replacing $\Sigma$ by the corresponding correlation matrix. Unless the entries of $A$ are already standardized causal effects, the matrix $A$ should be transformed under the same rescaling. However, regardless of standardization, the compatibility score is negative if and only if Assumption 2.4 is violated (see Figure 1 for an example). In the following sections, we provide evidence that true causal statements typically do not induce models violating Assumption 2.4. Hence, we view a *negative* compatibility score as evidence for *falsification* of the corresponding causal statements, while a *positive* compatibility score is *inconclusive*, since the causal statements could simply induce a plausible but wrong causal model.

### 2.6. Theoretical Analysis

**Plausibility of random causal models:** As a first justification for Assumption 2.4, we show that the expected compatibility score of true bivariate causal statements is positive for all distributions over causal models that satisfy only a few natural properties. We consider causal models given by linear SEMs with centered Gaussian noise. Such a model is fully specified by the strictly lower-triangular matrix $\Gamma$ of causal coefficients and a symmetric positive definite noise covariance matrix $\Sigma_{\mathbf{N}}$. Note that the covariance matrix $\Sigma_{\mathbf{X}}$ of the observable variables $\mathbf{X}$, that is needed to compute compatibility scores, is given by

$$\Sigma_{\mathbf{X}} = (I - \Gamma)^{-1} \Sigma_{\mathbf{N}} (I - \Gamma)^{-\top}.$$

**Assumption 2.8.** We consider probability distributions over $(\Gamma, \Sigma_{\mathbf{N}})$ satisfying the following properties:

1. (Unbiasedness) For all $i, j$, we have $\mathbb{E}[\Gamma_{ij}] = 0$;

2. (Independence of causal mechanisms [1]) The noise covariance matrix and all entries of $\Gamma$ are mutually independent.

3. (Non-degeneracy) $\Sigma_{\mathbf{N}} \succ 0$ almost surely and $\mathrm{Var}(\Gamma_{ij}) > 0$ for all $i > j$.

Note that these assumptions still allow for arbitrary choices of Gaussian noise, and arbitrary symmetric probability density functions for each causal coefficient.

**Theorem 2.9.** *For $n \geq 3$, let $(\Gamma, \Sigma_{\mathbf{N}})$ specify a random $n$-dimensional linear Gaussian SEM drawn from a distribution that satisfies Assumption 2.8. Define $A = (I - \Gamma)^{-1}$ to be the matrix of marginal bivariate causal effects. Then,*

$$\mathbb{E}_{(\Gamma, \Sigma_{\mathbf{N}})} \left[ \mathrm{comp}(\Sigma_{\mathbf{X}}, A) \right] > 0.$$

The proof of Theorem 2.9 decomposes $\mathrm{comp}(\Sigma_{\mathbf{X}}, A)$ into two components: a squared term, which is strictly positive by the non-degeneracy assumption, and a mixed product of terms whose expectation vanishes due to unbiasedness and independence of the causal coefficients. A full proof is given in Appendix B.

**Finite-sample estimation of the compatibility score:** In practice, we usually do not have access to the precise covariance matrix $\Sigma$ of the vector $\mathbf{X}$ of observed variables that is needed to compute the compatibility score of a list of causal statements $A$. Instead, we use samples $X^{(1)}, \ldots, X^{(N)}$ to compute an empirical covariance matrix

$$\widehat{\Sigma} = \frac{1}{N} \sum_{r=1}^{N} X^{(r)} X^{(r)\top}.$$

Let us define

$$V := \max_i \Sigma_{ii}, \qquad a := \max_{i<j} |A_{ji}|,$$

$$b := \max_{i<j} \sum_{k<i} \left| \sum_{\substack{P_1:k \rightsquigarrow i,\ P_2:k \rightsquigarrow j \\ P_1 \cap P_2 = \emptyset}} \Gamma^{P_1} \Gamma^{P_2} \right|,$$

where $\Gamma = I - A^{-1}$. Here, $a$ and $b$ are parameters that only depend on the matrix $A$ of bivariate causal statements. While $a$ simply captures the largest proclaimed total causal effect between two variables, the parameter $b$ essentially

[1]The independence of causal mechanisms assumption states that the conditional distributions of each variable given its parents should not be informative of each other (Peters et al., 2017). In our setting, this requires the rows of $\Gamma$ and the noise covariance matrix to be chosen independently. Our assumption is therefore a slight strengthening (see Gresele et al. (2021) for a similar proposal).

quantifies the largest proclaimed total contribution of causal back-door paths to the covariance between two variables. The following result states that polynomially many samples in $a, b, V$ and the number of variables $n$ suffice to approximate the true compatibility score well by using an empirical estimate of the covariance matrix. The proof is based on standard concentration results and can be found in Appendix B.

**Theorem 2.10.** *Fix $0 < \varepsilon, \delta < 1$, let $\mathbf{X}$ be an $n$-dimensional centered Gaussian vector with covariance matrix $\Sigma$ and suppose that $\widehat{\Sigma}$ is estimated from*

$$N \geq C \frac{n^4 (1 + a + b)^4 V^4}{\varepsilon^2} \log \frac{n}{\delta}$$

*many iid samples, where $C$ is a universal constant. Then, with probability at least $1 - \delta$, we have*

$$\left| \mathrm{comp}(\widehat{\Sigma}, A) - \mathrm{comp}(\Sigma, A) \right| \leq \epsilon.$$

### 2.7. Experiments

The source code for all our experiments can be found under https://github.com/ejahn17/compatibility-scores. First, we validate our approach using synthetic experiments. We sample random causal ground truth in the form of linear Gaussian SEMs with $n$ observed variables, $m$ hidden variables and sparsity parameter $p$, which sets causal coefficients to zero with probability $1 - p$ (see Appendix D.1). Starting from the true bivariate causal statements implied by the ground truth, we generate lists of bivariate statements of decreasing quality by adding independent Gaussian noise with increasing variance $\sigma$ to the causal coefficients. We then compute compatibility scores for these lists (Figure 2). Each data point in Figure 2 averages over 50 draws of different causal models and 20 draws of the random noise per model. Across all tested combinations of model parameters, the fraction of statements with positive compatibility scores strictly decreases as the error increases, which implies that our compatibility score successfully distinguishes between correct and incorrect statements on average. Our results also provide empirical evidence for the validity of Assumption 2.4, as most of the true sets of bivariate causal statements have positive compatibility scores. We further observe that the score is robust to variations in the number of hidden variables and that it becomes more successful at distinguishing between correct and incorrect causal statements as the number of observed variables and the density of the true causal model increases. This is consistent with the intuition that both factors increase the number of observed back-door paths in the multivariate model, which are unobserved in the bivariate marginals and therefore increase the difference of the confounding scores. We also examine the differences between compatibility scores that

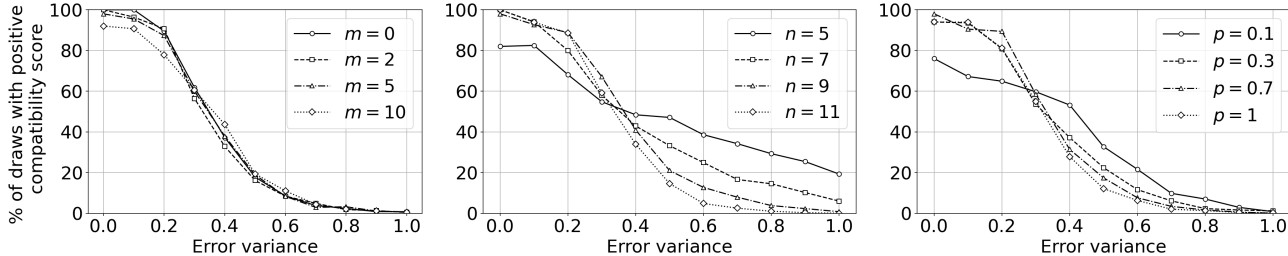

*Figure 2.* Percentage of positive compatibility scores for lists of bivariate causal statements on synthetic linear models with increasing amount of error. Each plot contains four different curves for four variations of a model parameter. The fixed model parameters in the first plot are $n = 10, p = 0.5$, in the second $m = 3, p = 0.5$, and in the third $n = 10, m = 3$.

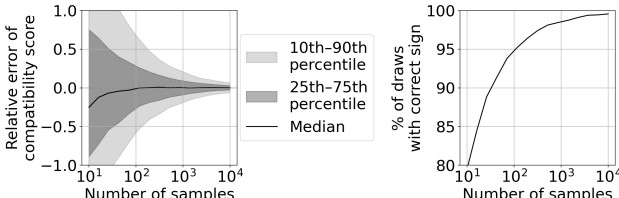

*Figure 3.* Relative error for empirical and true compatibility scores based on 150000 combinations of different model draws, sample draws and draws of random error in the causal statements with parameters $n = 10, p = 0.5, m = 3, \sigma = 0.2$.

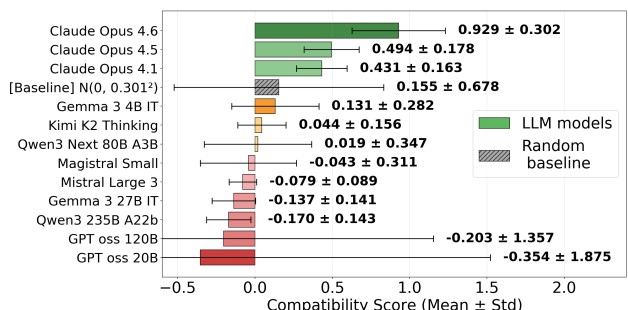

*Figure 4.* Compatibility scores for lists of bivariate statements obtained from different LLMs.

are computed with an empirical covariance matrix based on different numbers of samples and scores computed with the true covariance matrix (see Figure 3). The results suggest that the empirical scores converge faster to the true scores in practice than the worst-case analysis in Theorem 2.10 suggests. Moreover, less than 100 samples are sufficient for the signs of the empirical and the true scores to match in more than 90% of the test cases. See Appendix D.4 for additional plots for different parameter combinations.

In a second set of experiments, we compute compatibility scores for lists of linear bivariate causal statements produced by different large language models (LLMs) that we accessed via Amazon Bedrock. For the causal system, we consider seven variables from the gapminder dataset (Gapminder Foundation, 2026): *population density, literacy rate, average daily income, access to sanitation, percentage of adults that smoke, happiness score, life expectancy* (see Appendix D.2 for details). The dataset consists of country-level data points for 179 different countries over the years 1950-2025. We prompt LLMs to choose a causal ordering and estimate total linear causal effects for each pair of variables, based on variable descriptions and the empirical correlation matrix (see Appendix D.3). We compute compatibility scores averaged over 15 independent runs for each model and compare them to a random baseline, where causal coefficients are drawn from a centered normal distribution with variance matching the variance of the LLM outputs (Figure 4). The results show that our compatibility score

can demonstrate systematic differences across models, with higher-capacity models tending to achieve higher scores. Even though the random baseline scores positively in our experiment, many LLMs still receive negative scores, in which case our approach provides evidence for the incorrectness of the generated causal statements.

## 3. Incompatibility for Graphical Bivariate Causal Statements

### 3.1. Graphical Causal Statements

When quantitative causal statements are unavailable, it can still be desirable to have qualitative causal statements about the existence and direction of causal effects. In this section, we consider acyclic directed mixed graphs (ADMGs) as our causal models. An ADMG is a *mixed* graph on a set of variables $V$, that is, a graph that can have both directed edges and bidirected edges, with the restriction that the set of directed edges is acyclic. ADMGs have been widely studied as graphical causal models for causal structures with confounding (Pearl & Verma, 1995; Spirtes et al., 2001). Informally, a directed edge $v \rightarrow w$ represents a direct causal effect from $v$ to $w$, and a bidirected edge $v \leftrightarrow w$ indicates the existence of an unobserved causal backdoor path, i.e. confounding, between $v$ and $w$. The fact that ADMGs are capable of representing unobserved variables makes them

closed under marginalization.

**Definition 3.1.** Let $v, w$ be vertices in an ADMG $G$. A path between $v, w$ is a *confounding path* if both $v$ and $w$ are adjacent to an arrowhead of the path and no intermediate vertex is adjacent to two arrowheads (e.g. $v \leftrightarrow \rightarrow \rightarrow w$ or $v \leftarrow \leftrightarrow \rightarrow w$).

**Definition 3.2** (marginal ADMG, see Richardson et al. (2023)). Let $G$ be an ADMG on the vertex set $V \cup L$, where $V$ and $L$ are disjoint. The marginal ADMG $H$ on $V$ contains all edges of $G$ within $V$, and additionally

1. a directed edge $v \rightarrow w$ if there exists a directed path from $v$ to $w$ in $G$ with all intermediate vertices in $L$;

2. a bidirected edge $v \leftrightarrow w$ if there exists a confounding path from $v$ to $w$ in $G$ with all intermediate vertices in $L$.

This definition is motivated by the fact that if a normal distribution is *Markov* and *faithful* (see Appendix A) with respect to an ADMG $G$ on $V \cup L$, then its marginal distribution on $V$ is Markov and faithful with respect to the marginal ADMG $H$ on $V$ (see Corollaries 7.2 and 7.3 in Richardson & Spirtes (2002)). We now focus on evaluating lists of bivariate graphical causal statements.

**Definition 3.3** (bivariate statements and statement graph). A *bivariate graphical causal statement* for two variables $X_i, X_j$ specifies an ADMG on $X_i, X_j$, i.e. the existence and direction of a direct causal effect and the existence of confounding. Given a complete list of $\binom{n}{2}$ such statements for $n$ variables, we define the *statement graph* $\mathcal{G}$ to be the mixed graph obtained by taking the union of all bivariate ADMGs.

Faller et al. (2024) already introduced the concept of graphical compatibility for ADMGs.

**Definition 3.4** (graphical compatibility). A collection of ADMGs on subsets of a common vertex set $V$ is *graphically compatible* if they coincide with the marginalizations of a single large ADMG on $V$.

Specifically for complete lists of bivariate ADMGs, we have the following characterization of graphical compatibility:

**Lemma 3.5.** *A list of $\binom{n}{2}$ graphical bivariate causal statements with statement graph $\mathcal{G}$ is graphically compatible if and only if*

1. *the directed part of $\mathcal{G}$ is acyclic;*

2. *the directed part of $\mathcal{G}$ is transitively closed, i.e. if there is a directed path from $X_i$ to $X_j$, then there must be an edge from $X_i$ to $X_j$;*

3. *if there is a confounding path between $X_i$ and $X_j$, then there must be a bidirected edge between $X_i$ and $X_j$.*

The proof is straightforward and can be found in Appendix B.

### 3.2. Incompatibility Score

Let $d(\mathcal{G}, \mathcal{G}^*)$ denote the Hamming distance between two mixed graphs, that is the number of directed and bidirected edge deletions and additions needed to transform $\mathcal{G}$ into $\mathcal{G}^*$.

**Definition 3.6.** For a statement graph $\mathcal{G}$ summarizing a list of $\binom{n}{2}$ bivariate graphical causal statements, we define the *incompatibility score*

$$\mathrm{incomp}(\mathcal{G}) = \min_{\mathcal{G}^*} d(\mathcal{G}, \mathcal{G}^*),$$

where the minimum ranges over all mixed graphs $\mathcal{G}^*$ that satisfy the three properties stated in Lemma 3.5.

There is an important distinction from the graphical incompatibility score defined by Faller et al. (2024). While their paper assesses compatibility with a specific large ADMG, that is inferred by the same causal discovery algorithm as the respective small ADMGs, we do not assume access to an inferred multivariate ADMG. Instead, we implicitly optimize over the multivariate ADMG, by comparing $\mathcal{G}$ with the closest possible compatible set of statements. However, this generalization makes the problem of computing $\mathrm{incomp}(\mathcal{G})$ NP-hard. This is because it contains the following NP-hard problem.

ACYCLIC TRANSITIVITY EDITING: Given an acyclic directed graph $G$, find the minimum number of edge deletions and additions to make it transitively closed. See Weller et al. (2012) for a hardness proof.

Indeed, computing $\mathrm{incomp}(G)$ for an acyclic directed graph after adding all possible bidirected edges is equivalent to solving the transitivity editing problem. Still, this does not mean any attempt at computing $\mathrm{incomp}(\mathcal{G})$ is fruitless. Intuitively, the NP-hardness comes from regimes where $\mathrm{incomp}(\mathcal{G})$ is large. Indeed, a brute-force algorithm can decide in polynomial time if it is smaller than a given constant. But when $\mathrm{incomp}(\mathcal{G})$ is large, a precise computation is unnecessary for our purpose: any evidence for $\mathrm{incomp}(\mathcal{G})$ exceeding a certain threshold implies that the graph $\mathcal{G}$ is highly incompatible, which is already useful for falsification. Motivated by this insight, we present a heuristic algorithm for approximating $\mathrm{incomp}(\mathcal{G})$.

### 3.3. Heuristic Computation of the Incompatibility Score

The exact solution for the minimization problem in Definition 3.6 depends on the complex interaction between all three properties of Lemma 3.5. As a first heuristic, we decouple these constraints and address them sequentially. Given a mixed graph $\mathcal{G}$ with directed part $D = D(\mathcal{G})$, we proceed as follows:

STEP 1

First, we find a small set of directed edges whose deletion makes $D$ acyclic. This is exactly the MINIMUM FEEDBACK ARC problem, for which there is a heuristic algorithm called GreedyFAS, which runs in linear time and has good empirical performance (Eades et al., 1993; Simpson et al., 2016). The algorithm iteratively constructs a vertex ordering by picking sources and sinks whenever possible, otherwise, it greedily picks the vertex with maximal difference between out- and in-degree instead of a source. Its output set $E_{\text{cycles}}$ is the set of edges that is not consistent with the obtained ordering (see Appendix C for more details).

STEP 2

Set $D' = D \setminus E_{\text{cycles}}$ to be the directed acyclic graph obtained in step 1. We design a greedy, heuristic algorithm to solve the ACYCLIC TRANSITIVITY EDITING problem for $D'$. Let the *transitive closure $tc(D')$* be the smallest directed graph that contains $D'$ and is transitively closed. Our algorithm GreedyTE iteratively deletes the edge that reduces the Hamming distance from the current graph to its transitive closure as much as possible, resulting in the deletion of $E_{\text{del}} \subseteq E(D')$. Once no more reduction is possible, the algorithm adds the set of edges $E_{\text{add}}$ that is missing to get to $tc(D' \setminus E_{\text{del}})$ (see Appendix C).

STEP 3

After steps 1 and 2, the directed part $D$ of the statement graph $\mathcal{G}$ has been transformed into a DAG $D' = (D \setminus (E_{\text{cycles}} \cup E_{\text{del}})) \cup E_{\text{add}}$ that is transitively closed. Now, we keep $D'$ fixed and only operate on the bidirected edges of $\mathcal{G}$. Define the *confounding path closure $cpc(G)$* of a mixed graph $G$ to be the graph obtained from $G$ by adding bidirected edges between each pair of vertices that is connected by a confounding path. We then apply a greedy procedure analogous to Step 2, where the transitive closure is replaced by $cpc(G)$, to identify bidirected edge sets $B_{\text{del}}$ and $B_{\text{add}}$ whose deletion and addition makes the resulting graph satisfy property 3 of Lemma 3.5. Again, the full algorithm is given in Appendix C.

We report

$$c(\mathcal{G}) := |E_{\text{cycles}}| + |E_{\text{add}}| + |E_{\text{del}}| + |B_{\text{add}}| + |B_{\text{del}}| \tag{5}$$

as a heuristic approximation for $\text{incomp}(\mathcal{G})$. This approximation satisfies the following properties:

**Lemma 3.7.** *For any statement graph $\mathcal{G}$, we have*

1. $c(\mathcal{G}) \geq \text{incomp}(\mathcal{G})$;

2. $c(\mathcal{G}) = 0 \iff \text{incomp}(\mathcal{G}) = 0$.

Hence, whenever $\text{incomp}(\mathcal{G})$ is either zero or it is large, this reflects correctly in our heuristic approximation. We give a proof of Lemma 3.7 in Appendix B.

### 3.4. Experiments

Again, we start with synthetic experiments and sample causal ground truth with $n$ observed variables, $m$ hidden variables and sparsity parameter $p$ (see Appendix D.1). For each list of correct bivariate causal statements, we inject a fixed number of random errors, by randomly selecting an edge out of all possible directed and bidirected edges and deleting or inserting it, depending on whether it was present or absent. Then we compute incompatibility scores using the heuristic approach of Section 3.3, see Figure 5. As expected, incompatibility scores increase on average monotonically with the number of injected errors. This suggests that the incompatibility score can serve as a tool for comparing the quality of different lists of graphical bivariate causal statements for the same causal graph. For small or sparse graphs, the scores also approximate the true numbers of errors quite closely on average, and they are robust against variations in the number of hidden confounders. However, for larger and denser models, the scores tend to overestimate the true error count, which indicates that the approximation quality of our heuristic algorithm decreases as the number of injected errors is an upper bound for the exact incompatibility score.

We also apply our approach to graphical bivariate causal statements obtained from LLMs, using the same dataset and models as in Section 2.7 (see Appendix D.3 for prompt details). The results are shown in Figure 6. Among all derived statement graphs, only one achieves $\text{incomp}(\mathcal{G}) = 0$, and it is complete, meaning that it only needs to be acyclic to be compatible. Indeed, several low-scoring LLM outputs are extremely dense statement graphs, which achieve low incompatibility scores, but are not very informative[2]. This is why we also report incompatibility scores for statement graphs filtered by a capped edge density, see Figure 7. Among this subset of more informative statement graphs, the results again reveal a correlation between low incompatibility scores and higher general model capacity. We also provide a more detailed breakdown of incompatibility scores in Appendix D.4.

### 4. Discussion

We developed two complementary approaches to assess bivariate causal statements based on their mutual compatibility. For graphical statements, we approximate the minimum

---

[2]For this reason, the quality of a set of causal statements should never be judged based on compatibility alone, but also account for informativeness and falsifiability (Popper, 1959). Clearly, stating an unconfounded causal link is a more specific statement - and thus more falsifiable - than allowing confounding in addition.

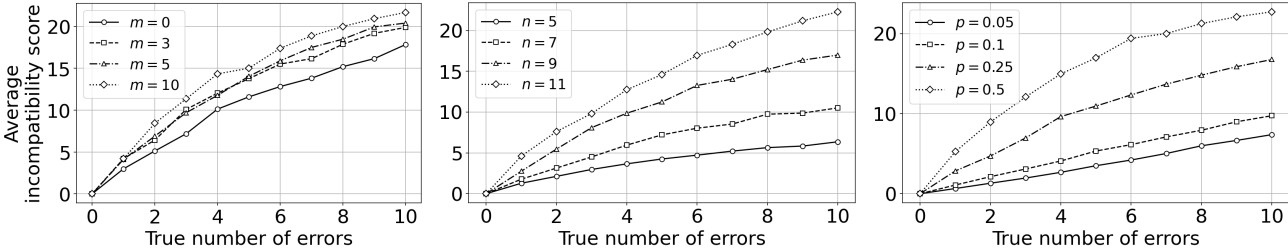

*Figure 5.* Average incompatibility scores for lists of bivariate causal statements on synthetic graphical models with increasing numbers of errors. The fixed parameters in the first plot are $n = 10$, $p = 0.3$, in the second $m = 3$, $p = 0.3$, and in the third $n = 10$, $m = 3$.

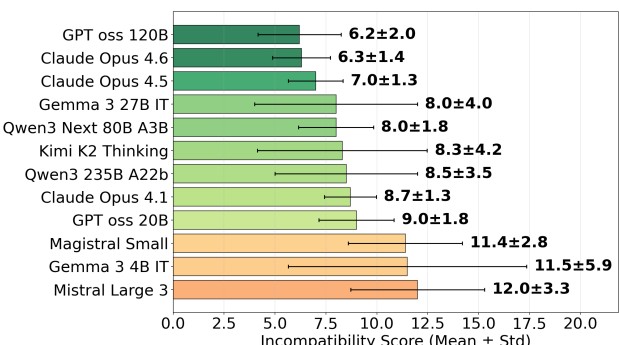

*Figure 6.* Incompatibility scores of statement graphs derived from LLMs, averaged over 10 repetitions of the same query.

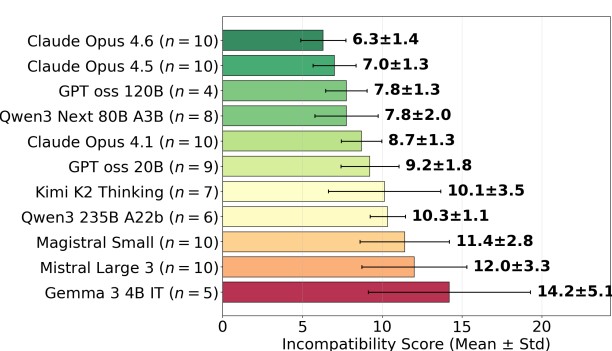

*Figure 7.* For each LLM, we report the size $n$ of the subset of statement graphs that have edge density at most $2/3$ (out of 10 statement graphs in total), and plot average incompatibility scores for this subset.

number of statements that need to change to give rise to a single consistent multivariate causal graph under the assumptions of acyclicity and faithfulness. In the absence of faithfulness and when quantitative linear causal statements are available, we measure compatibility by evaluating how well the unique induced linear multivariate model satisfies our plausibility criterion, namely that it should exhibit less confounding than its bivariate marginals. Importantly, a positive compatibility score does not imply correctness of the causal statements, but a negative score provides evidence for internal inconsistencies to some degree.

There is potential for future work that extends our framework to cases where the collection of bivariate causal statements may be incomplete, or accompanied by statements over small subsets of variables. Especially graphical compatibility seems amenable to such a generalization, while extending compatibility for linear statements may require new ideas, as the existence of a unique composite multivariate model can no longer be guaranteed.

Another interesting direction would be to test the application of our compatibility scores for causal inference in the setting where incomplete lists of bivariate causal statements are already available. The missing statements could then be obtained by maximizing the compatibility score (or minimizing the incompatibility score).

Finally, compatibility scores can provide a meaningful quantitative comparison between collections of causal claims in the absence of causal ground truth. Thus, our methods could guide the automated evaluation of causal statements, for example as part of post-training, filtering, or validation pipelines for LLMs. This could be a step towards obtaining more reliable causal information from AI models that can effectively support human decision-making.

## Acknowledgements

E.J. was supported by the National Science Foundation under Grant No. CCF-2321079.

## Impact Statement

This paper describes an approach for assessing the reliability of AI-generated causal statements. Our goal is to enable future improvements of their accuracy. To mitigate the risk of over-interpreting compatibility scores, we want to emphasize that a high score does not provide a proof of correctness of the respective causal statements, it only indicates that they pass a sanity check.

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

# A. Additional Background

Below, we summarize how causal graphical models such as ADMGs can be linked to probability distributions (see Richardson & Spirtes (2002)):

**Definition A.1** (Collider). Given a path in an ADMG $G$, a *collider* is an interior vertex $x$ on the path such that the edges preceding and succeeding $x$ have arrowheads pointing to $x$ (that is, $\rightarrow x \leftarrow$ or $\leftrightarrow x \leftarrow$ or $\rightarrow x \leftrightarrow$ or $\leftrightarrow x \leftrightarrow$).

**Definition A.2** ($m$-separation). Let $G$ be an ADMG with disjoint vertex sets $X, Y, Z$. Then, $X$ and $Y$ are said to be $m$-connected given $Z$ if there exists a path connecting a vertex in $X$ with a vertex in $Y$ such that

1. each non-collider on the path is not in $Z$;

2. for each collider on the path, there exists a directed path to a vertex in $Z$.

If $X$ and $Y$ are not $m$-connected given $Z$, they are said to be $m$-separated given $Z$.

**Definition A.3** (Markov condition). A probability distribution $P$ on $V$ satisfies the *Markov condition* with respect to an ADMG $G$ with vertex set $V$ if for any disjoint subsets $X, Y, Z$, we have

$$X \text{ and } Y \text{ are } m\text{-separated given } Z \text{ in } G \implies X \text{ and } Y \text{ are independent conditioned on } Z \text{ in } P.$$

**Definition A.4** (Faithfulness condition). A probability distribution $P$ on $V$ satisfies the *faithfulness condition* with respect to an ADMG $G$ with vertex set $V$ if for any disjoint subsets $X, Y, Z$, we have

$$X \text{ and } Y \text{ are independent conditioned on } Z \text{ in } P \implies X \text{ and } Y \text{ are } m\text{-separated given } Z \text{ in } G.$$

The Markov and faithfulness conditions are standard assumptions for the task of inferring a causal graph from an empirical probability distribution. For linear SEMs $\mathbf{X} = \Gamma\mathbf{X} + \mathbf{N}$ with Gaussian noise, one can show that the distribution of $\mathbf{X}$ always satisfies the Markov condition with respect to the corresponding ADMG on $\mathbf{X}$ that is defined by setting $X_i \rightarrow X_j$ if and only if $\Gamma_{ji} \neq 0$ and $X_i \leftrightarrow X_j$ if and only if $\text{Cov}(N_i, N_j) \neq 0$. In contrast, the faithfulness condition is not always satisfied, even for linear Gaussian models (which is the case when the contributions of multiple $m$-connecting paths to the covariance between two variables exactly cancel), and we do not need this condition for our approach developed in Section 2. However, the faithfulness condition is needed to ensure that a causal graph inferred from a marginal probability distribution coincides with the marginal causal graph that is inferred from the full probability distribution. The soundness of Definition 3.4 relies on this fact, and therefore our approach in Section 3 implicitly relies on the faithfulness assumption.

# B. Proofs

**Lemma 2.3** (Existence of a unique compatible SEM). *Let $A$ be a unit lower-triangular matrix of linear bivariate causal statements for a vector $\mathbf{X}$ of observed variables and define $\Gamma = I - A^{-1}$. Then, the multivariate structural equation model*

$$\mathbf{X} = \Gamma\mathbf{X} + \mathbf{N},$$

*where the distribution of $\mathbf{N}$ is defined as the distribution of $(I - \Gamma)\mathbf{X}$, is the unique linear SEM whose pairwise marginal submodels are*

$$X_j = A_{ji}X_i + \tilde{N}_{ij}.$$

*Proof.* Consider an arbitrary SEM $\mathbf{X} = \Gamma\mathbf{X} + \mathbf{N}$. By Lemma 2.1, marginalizing to the pair $(X_i, X_j)$ with $i < j$ results in the SEM

$$\begin{pmatrix} X_i \\ X_j \end{pmatrix} = (\Gamma_{YY} + \Gamma_{YZ}(I - \Gamma_{ZZ})^{-1}\Gamma_{ZY}) \begin{pmatrix} X_i \\ X_j \end{pmatrix} + \tilde{\mathbf{N}},$$

where $Y = i, j$ and $Z = \{1, \ldots, n\} \setminus \{i, j\}$. Since $\Gamma$ is strictly lower-triangular, we have that

$$\Gamma_{YY} + \Gamma_{YZ}(I - \Gamma_{ZZ})^{-1}\Gamma_{ZY} = \begin{pmatrix} 0 & 0 \\ r & 0 \end{pmatrix},$$

for some entry $r \in \mathbb{R}$. On the other hand, block matrix inversion of $(I - \Gamma)$ using the Schur complement tells us that

$$(I - \Gamma)_{YY}^{-1} = (I - \Gamma_{YY} - \Gamma_{YZ}(I - \Gamma_{ZZ})^{-1}\Gamma_{ZY})^{-1} = \begin{pmatrix} 1 & 0 \\ -r & 1 \end{pmatrix}^{-1} = \begin{pmatrix} 1 & 0 \\ r & 1 \end{pmatrix}.$$

This implies $r = (I - \Gamma)_{ji}^{-1}$, so the pairwise marginal submodels of an SEM with causal coefficients $\Gamma$ are given by

$$X_j = (I - \Gamma)_{ji}^{-1} X_i + \tilde{N}_{ij}.$$

Requiring that all these submodels align with $X_j = \alpha_{ij} X_i + \tilde{N}_{ij}$ is therefore equivalent to $(I - \Gamma)^{-1} = A$ or $\Gamma = I - A^{-1}$, which completes the proof. $\qquad \square$

**Theorem 2.9.** *For $n \geq 3$, let $(\Gamma, \Sigma_{\mathbf{N}})$ specify a random $n$-dimensional linear Gaussian SEM drawn from a distribution that satisfies Assumption 2.8. Define $A = (I - \Gamma)^{-1}$ to be the matrix of marginal bivariate causal effects. Then,*

$$\mathbb{E}_{(\Gamma, \Sigma_{\mathbf{N}})} [\mathrm{comp}(\Sigma_{\mathbf{X}}, A)] > 0.$$

*Proof.* For $1 \leq i < j \leq n$, we define

$$B_{ij} := \sum_{k < i} \Sigma_{\mathbf{X}, kk} \cdot \sum_{\substack{P_1 : k \rightsquigarrow i \\ P_2 : k \rightsquigarrow j \\ P_1 \cap P_2 = \emptyset}} \Gamma^{P_1} \Gamma^{P_2}$$

to be the part of the covariance between $X_i, X_j$ coming from back-door paths through observed variables and

$$\varepsilon_{ij} := \sum_{\ell \neq k \leq j} \Sigma_{\mathbf{N}, lk} \cdot \sum_{\substack{P_1 : \ell \rightsquigarrow i \\ P_2 : k \rightsquigarrow j \\ P_1 \cap P_2 = \emptyset}} \Gamma^{P_1} \Gamma^{P_2}$$

to be the part of the covariance between $X_i, X_j$ coming from back-door paths that are not fully observed. First, we show that $B_{ij}$ and $\varepsilon_{ij}$ are uncorrelated. For $i = 1$, we have $B_{ij} = 0$, so this is trivially true. For $i > 1$, we get

$$\mathbb{E}[B_{ij}\varepsilon_{ij}] = \sum_{k < i} \sum_{\ell \neq m \leq j} \sum_{\substack{P_1 : k \rightsquigarrow i \\ P_2 : k \rightsquigarrow j \\ P_1 \cap P_2 = \emptyset}} \sum_{\substack{Q_1 : \ell \rightsquigarrow i \\ Q_2 : m \rightsquigarrow j \\ Q_1 \cap Q_2 = \emptyset}} \mathbb{E} \left[ \Sigma_{\mathbf{X}, kk} \cdot \Sigma_{\mathbf{N}, lm} \cdot \Gamma^{P_1} \Gamma^{P_2} \Gamma^{Q_1} \Gamma^{Q_2} \right] \qquad (6)$$

Fix a summand in the expression above, determined by specific choices of $k, \ell, m, P_1, P_2, Q_1, Q_2$. Since $Q_1, Q_2$ are disjoint paths with all different endpoints, they cannot fully contain the paths $P_1, P_2$ as those intersect in the common endpoint $k$. Hence, there must exist an entry $\Gamma_{rs}$ with $r > k$ that only appears once in the product $\Gamma^{P_1} \Gamma^{P_2} \Gamma^{Q_1} \Gamma^{Q_2}$. Note that $\Sigma_{\mathbf{X}, kk}$ only depends on $\Sigma_{\mathbf{N}}$ and entries $\Gamma_{pq}$ with $p \leq k$. By part (2) of Assumption 2.8, the entry $\Gamma_{rs}$ is therefore independent from all other factors in the product $\Sigma_{\mathbf{X}, kk} \cdot \Sigma_{\mathbf{N}, lm} \cdot \Gamma^{P_1} \Gamma^{P_2} \Gamma^{Q_1} \Gamma^{Q_2}$. This implies

$$\mathbb{E} \left[ \Sigma_{\mathbf{X}, kk} \cdot \Sigma_{\mathbf{N}, lm} \cdot \Gamma^{P_1} \Gamma^{P_2} \Gamma^{Q_1} \Gamma^{Q_2} \right] = \mathbb{E} \left[ \Gamma_{rs} \right] \cdot \mathbb{E} \left[ \Sigma_{\mathbf{X}, kk} \cdot \Sigma_{\mathbf{N}, lm} \cdot \Gamma^{P_1} \Gamma^{P_2} \Gamma^{Q_1} \Gamma^{Q_2} \cdot \frac{1}{\Gamma_{rs}} \right] = 0,$$

by part (1) of Assumption 2.8. Since this argument holds for all summands of equation (6), we get

$$\mathbb{E}[B_{ij}\varepsilon_{ij}] = 0.$$

Now, recall equation (4) stating that

$$\Sigma_{\mathbf{X}, ij} = \Sigma_{\mathbf{X}, ii} \cdot A_{ji} + B_{ij} + \varepsilon_{ij}.$$

Therefore, we get for the bivariate confounding between $X_i$ and $X_j$

$$\mathcal{C}_{ij}^{\mathrm{biv}}(\Sigma_{\mathbf{X}}, A_{ji}) = (B_{ij} + \varepsilon_{ij})^2$$

and the multivariate confounding is given by

$$\mathcal{C}_{ij}^{\mathrm{mult}}(\Sigma_{\mathbf{X}}, \Gamma) = \varepsilon_{ij}^2.$$

Hence, we can write

$$\mathbb{E}[\mathrm{comp}(\Sigma_{\mathbf{X}}, A)] = \mathbb{E}\left[\sum_{i<j}(B_{ij} + \varepsilon_{ij})^2 - \varepsilon_{ij}^2\right] = \sum_{i<j}\left(\mathbb{E}\left[B_{ij}^2\right] + 2\,\mathbb{E}\left[B_{ij}\varepsilon_{ij}\right]\right) = \sum_{i<j}\mathbb{E}[B_{ij}^2] \geq 0.$$

For $i = 2, j = 3$, we get from part (2) of Assumption 2.8

$$\mathbb{E}[B_{23}^2] = \mathbb{E}[\Sigma_{\mathbf{X},11}^2 \cdot \Gamma_{21}^2 \cdot \Gamma_{31}^2] = \mathbb{E}[\Sigma_{\mathbf{N},11}^2 \cdot \Gamma_{21}^2 \cdot \Gamma_{31}^2] = \mathbb{E}[\Sigma_{\mathbf{N},11}^2] \cdot \mathbb{E}[\Gamma_{21}^2] \cdot \mathbb{E}[\Gamma_{31}^2].$$

By part (3) of Assumption 2.8, we have that $\Sigma_{\mathbf{N},11}^2 > 0$ almost surely, and $\mathbb{E}[\Gamma_{21}^2] = \mathrm{Var}(\Gamma_{21}) > 0, \mathbb{E}[\Gamma_{31}^2] = \mathrm{Var}(\Gamma_{31}) > 0$. This implies $\mathbb{E}[B_{23}^2] > 0$ and therefore also

$$\mathbb{E}[\mathrm{comp}(\Sigma_{\mathbf{X}}, A)] > 0.$$

□

**Theorem 2.10.** *Fix $0 < \varepsilon, \delta < 1$, let $\mathbf{X}$ be an $n$-dimensional centered Gaussian vector with covariance matrix $\Sigma$ and suppose that $\widehat{\Sigma}$ is estimated from*

$$N \geq C\frac{n^4(1 + a + b)^4 V^4}{\varepsilon^2}\log\frac{n}{\delta}$$

*many iid samples, where $C$ is a universal constant. Then, with probability at least $1 - \delta$, we have*

$$\left|\mathrm{comp}(\widehat{\Sigma}, A) - \mathrm{comp}(\Sigma, A)\right| \leq \epsilon.$$

*Proof.* Since $X$ is centered Gaussian and $\max_i \Sigma_{ii} \leq V$, a standard concentration result (see Ravikumar et al. (2011), Lemma 1) shows that there exists a universal constant $c > 0$ such that, for all $0 < \gamma \leq V$,

$$\Pr\left[\max_{p,q}|\widehat{\Sigma}_{pq} - \Sigma_{pq}| > \gamma\right] \leq 4n^2 \exp\left(-cN\frac{\gamma^2}{V^2}\right).$$

Let $\mathcal{E}_\gamma$ denote the event

$$\mathcal{E}_\gamma := \left\{\max_{p,q}|\widehat{\Sigma}_{pq} - \Sigma_{pq}| \leq \gamma\right\}.$$

Conditional on $\mathcal{E}_\gamma$, we prove a bound on $|\mathrm{comp}(\widehat{\Sigma}, A) - \mathrm{comp}(\Sigma, A)|$. For $i < j$ and fixed $A$, define the bivariate residual

$$B_{ij}(\Sigma) := \Sigma_{ij} - A_{ji}\Sigma_{ii},$$

and the multivariate residual

$$M_{ij}(\Sigma) := \Sigma_{ij} - \sum_{k \leq i}\Sigma_{kk}\sum_{\substack{P_1:k \rightsquigarrow i,\ P_2:k \rightsquigarrow j \\ P_1 \cap P_2 = \emptyset}}\Gamma^{P_1}\Gamma^{P_2},$$

where $\Gamma = I - A^{-1}$. The term corresponding to $k = i$ equals $A_{ji}\Sigma_{ii}$, because it sums over all directed paths from $i$ to $j$. Hence

$$M_{ij}(\Sigma) = \Sigma_{ij} - A_{ji}\Sigma_{ii} - \sum_{k < i}\Sigma_{kk}\sum_{\substack{P_1:k \rightsquigarrow i,\ P_2:k \rightsquigarrow j \\ P_1 \cap P_2 = \emptyset}}\Gamma^{P_1}\Gamma^{P_2}.$$

Let $K := 1 + a + b$ with $a, b$ as defined in Section 2.6. Then, $B_{ij}$ and $M_{ij}$ are linear functions of the entries of $\Sigma$ with coefficient $\ell_1$-norm at most $K$. On the event $\mathcal{E}_\gamma$, this implies

$$|B_{ij}(\widehat{\Sigma}) - B_{ij}(\Sigma)| \leq K\gamma, \qquad |M_{ij}(\widehat{\Sigma}) - M_{ij}(\Sigma)| \leq K\gamma.$$

Since $|\Sigma_{pq}| \leq V$ for all $p, q$ by Cauchy–Schwarz, we also have

$$|B_{ij}(\Sigma)|, \ |M_{ij}(\Sigma)| \leq KV,$$

and on $\mathcal{E}_\gamma$, assuming $\gamma \leq V$, we have $|\widehat{\Sigma}_{pq}| \leq 2V$, and therefore

$$|B_{ij}(\widehat{\Sigma})|, \ |M_{ij}(\widehat{\Sigma})| \leq 2KV.$$

Using $|x^2 - y^2| = |x - y||x + y|$, we get

$$\left| \mathcal{C}_{ij}^{\text{biv}}(\widehat{\Sigma}, A_{ji}) - \mathcal{C}_{ij}^{\text{biv}}(\Sigma, A_{ji}) \right| = \left| B_{ij}(\widehat{\Sigma})^2 - B_{ij}(\Sigma)^2 \right| \leq K\gamma \cdot 3KV = 3K^2V\gamma,$$

and similarly,

$$\left| \mathcal{C}_{ij}^{\text{mult}}(\widehat{\Sigma}, \Gamma) - \mathcal{C}_{ij}^{\text{mult}}(\Sigma, \Gamma) \right| = \left| M_{ij}(\widehat{\Sigma})^2 - M_{ij}(\Sigma)^2 \right| \leq 3K^2V\gamma.$$

Thus, for each pair $i < j$, we obtain

$$\left| \left( \mathcal{C}_{ij}^{\text{biv}}(\widehat{\Sigma}, A_{ji}) - \mathcal{C}_{ij}^{\text{mult}}(\widehat{\Sigma}, \Gamma) \right) - \left( \mathcal{C}_{ij}^{\text{biv}}(\Sigma, A_{ji}) - \mathcal{C}_{ij}^{\text{mult}}(\Sigma, \Gamma) \right) \right| \leq 6K^2V\gamma.$$

Summing over at most $n^2/2$ pairs gives

$$\left| \text{comp}(\widehat{\Sigma}, A) - \text{comp}(\Sigma, A) \right| \leq 3n^2K^2V\gamma = \varepsilon,$$

where the last step follows by choosing $\gamma = \frac{\epsilon}{3n^2K^2V}$. Finally, the Gaussian covariance concentration bound shows that $\mathcal{E}_\gamma$ holds with probability at least $1 - \delta$ whenever

$$N \geq C' \frac{V^2}{\gamma^2} \log \frac{n}{\delta} = C \frac{n^4 K^4 V^4}{\varepsilon^2} \log \frac{n}{\delta}$$

for some sufficiently large constant $C$. This completes the proof. $\qquad\square$

**Lemma 3.5.** *A list of $\binom{n}{2}$ graphical bivariate causal statements with statement graph $\mathcal{G}$ is graphically compatible if and only if*

1. *the directed part of $\mathcal{G}$ is acyclic;*

2. *the directed part of $\mathcal{G}$ is transitively closed, i.e. if there is a directed path from $X_i$ to $X_j$, then there must be an edge from $X_i$ to $X_j$;*

3. *if there is a confounding path between $X_i$ and $X_j$, then there must be a bidirected edge between $X_i$ and $X_j$.*

*Proof.* First, suppose the statement graph $\mathcal{G}$ satisfies the three conditions of Lemma 3.5. By the first condition, $\mathcal{G}$ is an ADMG itself. By the second condition, all its directed edges coincide with the directed edges of its pairwise marginalizations and by the third condition, all its bidirected edges coincide with the bidirected edges of its pairwise marginalizations. Hence, $\mathcal{G}$ itself is a multivariate ADMG that is compatible with all bivariate statements.

Secondly, suppose the statement graph $\mathcal{G}$ is graphically compatible, so there exists a multivariate ADMG $G$ whose pairwise marginalizations coincide with the edges of $\mathcal{G}$. Assume that all vertices are ordered consistently with respect to the directed edges of $G$. Then, also the directed edges of the pairwise marginalizations of $G$ (which are obtained from directed paths), must be consistent with the vertex ordering. Hence, the directed part of $\mathcal{G}$ is acyclic, showing that the first condition of Lemma 3.5 holds. Next, suppose there is a directed path from $X_i$ to $X_j$ in $\mathcal{G}$. Every directed edge $X_k \to X_\ell$ in the path

corresponds to either a directed edge $X_k \to X_\ell$ in $G$ or a directed path from $X_k$ to $X_\ell$. Concatenating all these directed paths gives a directed path from $X_i$ to $X_j$ in $G$. Hence, marginalizing $G$ onto the vertex set $\{X_i, X_j\}$ results in an edge $X_i \to X_j$, which proves that $\mathcal{G}$ satisfies the second property of Lemma 3.5. Finally, assume that there is a confounding path between $X_i, X_j$ in $\mathcal{G}$. As before, each directed edge on this path must correspond to a directed path in $G$ and if there is a bidirected edge on the path, then it must correspond to a confounding path in $G$. Concatenating these directed paths and the confounding path, gives a confounding path between $X_i$ and $X_j$ in $G$, which implies the existence of a bidirected edge $X_i \leftrightarrow X_j$ in the pairwise marginal ADMG. This shows that $\mathcal{G}$ also satisfies the third property of Lemma 3.5 and completes the proof. □

**Lemma 3.7.** *For any statement graph $\mathcal{G}$, we have*

1. $c(\mathcal{G}) \geq \mathrm{incomp}(\mathcal{G})$ ;

2. $c(\mathcal{G}) = 0 \iff \mathrm{incomp}(\mathcal{G}) = 0$.

*Proof.* For part (1) of Lemma 3.7, it suffices to show that $\mathcal{G}^* = \mathcal{G} \setminus (E_{\mathrm{cycles}} \cup E_{\mathrm{del}} \cup B_{\mathrm{del}}) \cup E_{\mathrm{add}} \cup B_{\mathrm{add}}$ is graphically compatible. Let $\mathcal{G}' := \mathcal{G} \setminus (E_{\mathrm{cycles}} \cup E_{\mathrm{del}})$. By definition, $B_{\mathrm{add}}$ completes the confounding path closure of $\mathcal{G}' \setminus B_{\mathrm{del}} \cup E_{\mathrm{add}}$. Hence, $\mathcal{G}^*$ satisfies property 3 of Lemma 3.5. Similarly, by definition of $E_{\mathrm{add}}$, the graph $\mathcal{G}' \cup E_{\mathrm{add}}$ is the directed transitive closure of $\mathcal{G}'$. This property cannot be destroyed by adding or deleting bidirected edges, which implies that $\mathcal{G}^*$ is also transitively closed. Finally, we know that by definition of $E_{\mathrm{cycles}}$, the graph $\mathcal{G} \setminus E_{\mathrm{cycles}}$ is acyclic. This implies that $\mathcal{G}'$ is also acyclic. Since the directed transitive closure of an acyclic graph is still acyclic, we know that $\mathcal{G}' \cup E_{\mathrm{add}}$ is acyclic. Again, acyclicity cannot be destroyed by adding or removing bidirected edges, which implies that $\mathcal{G}^*$ is acyclic.

To show part 2 of Lemma 3.7, note that whenever $\mathrm{incomp}(\mathcal{G}) = 0$, i.e. $\mathcal{G}$ is graphically compatible, we must have $E_{\mathrm{cycles}} = E_{\mathrm{add}} = E_{\mathrm{del}} = B_{\mathrm{add}} = B_{\mathrm{del}} = \emptyset$, and hence $c(\mathcal{G}) = 0$. The converse follows from part 1. □

# C. Algorithms

In the description of Algorithm 1, $d_H^{\mathrm{in}}(v)$ denotes the in-degree of a vertex $v$ with respect to the graph $H$ and similarly, $d_H^{\mathrm{out}}(v)$ denotes the out-degree. In the description of Algorithm 2, $tc(H)$ denotes the transitive closure of $H$ and in the description of Algorithm 3, $cpc(H)$ denotes the confounding path closure of $H$ (see Section 3.3).

---

**Algorithm 1** Greedy Feedback Arc Set (GreedyFAS, Eades et al. (1993))

---

**Input:** Directed graph $D = (V, E)$
**Output:** Feedback arc set $E_{\mathrm{cycles}} \subseteq E$
Initialize empty lists $L \leftarrow [\,]$, $R \leftarrow [\,]$
Initialize $H \leftarrow D$
**while** $V(H) \neq \emptyset$ **do**
  **if** $H$ has a source $v$ (i.e. $d_H^{\mathrm{in}}(v) = 0$) **then**
    Remove $v$ from $H$
    Append $v$ to the end of $L$
  **else if** $H$ has a sink $v$ (i.e. $d_H^{\mathrm{out}}(v) = 0$) **then**
    Remove $v$ from $H$
    Prepend $v$ to the beginning of $R$
  **else**
    Choose $v \in V(H)$ maximizing $d_H^{\mathrm{out}}(v) - d_H^{\mathrm{in}}(v)$
    Remove $v$ from $H$
    Append $v$ to the end of $L$
  **end if**
**end while**
Let $\pi$ be the linear ordering given by concatenation $L \circ R$
$E_{\mathrm{cycles}} \leftarrow \{(u \to v) \in E : \pi(u) > \pi(v)\}$
**return** $E_{\mathrm{cycles}}$

---

---

**Algorithm 2** Greedy Transitivity Editing (GreedyTE)

---

**Input:** Acyclic directed graph $D = (V, E)$
**Output:** Edge sets $E_{\text{del}}$ and $E_{\text{add}}$ such that $D' = (V, (E \cup E_{\text{add}}) \setminus E_{\text{del}})$ is transitively closed
Initialize $E_{\text{del}} \leftarrow \emptyset$
Initialize $H \leftarrow D$
**repeat**
   Initialize $bestGain \leftarrow 0$
   Initialize $e^{\star} \leftarrow \emptyset$
   **for each** edge $e \in E(H)$ **do**
     Compute $gain \leftarrow |tc(H)| - |tc(H \setminus e)| - 1$
     **if** $gain > bestGain$ **then**
       $bestGain \leftarrow gain$
       $e^{\star} \leftarrow e$
     **end if**
   **end for**
   **if** $bestGain > 0$ **then**
     $E_{\text{del}} \leftarrow E_{\text{del}} \cup \{e^{\star}\}$
     $H \leftarrow H \setminus e^{\star}$
   **end if**
**until** $bestGain = 0$
Define $D' \leftarrow D \setminus E_{\text{del}}$
Set $E_{\text{add}} \leftarrow E(tc(D')) \setminus E(D')$
**return** $(E_{\text{del}}, E_{\text{add}})$

---

**Algorithm 3** Greedy Confounding Path Closure (GreedyCPC)

---

**Input:** Mixed graph $G$ with bidirected edge set $B = B(G)$
**Output:** Bidirected edge sets $B_{\text{del}} \subseteq B$ and $B_{\text{add}}$ such that deleting $B_{\text{del}}$ from $G$ and adding $B_{\text{add}}$ makes it equal to its confounding path closure
Initialize $B_{\text{del}} \leftarrow \emptyset$
Initialize $H \leftarrow G$
**repeat**
   Initialize $bestGain \leftarrow 0$
   Initialize $e^{\star} \leftarrow \emptyset$
   **for each** edge $e \in B(H)$ **do**
     Compute $gain \leftarrow |cpc(H)| - |cpc(H \setminus e)| - 1$
     **if** $gain > bestGain$ **then**
       $bestGain \leftarrow gain$
       $e^{\star} \leftarrow e$
     **end if**
   **end for**
   **if** $bestGain > 0$ **then**
     $B_{\text{del}} \leftarrow B_{\text{del}} \cup \{e^{\star}\}$
     $H \leftarrow H \setminus e^{\star}$
   **end if**
**until** $bestGain = 0$
Define $G' \leftarrow G \setminus B_{\text{del}}$
Set $B_{\text{add}} \leftarrow B(cpc(G')) \setminus B(G')$
**return** $(B_{\text{del}}, B_{\text{add}})$

---

## D. Experiment Details

### D.1. Synthetic Model Generation

We begin by stating the details for sampling causal ground truth in our synthetic experiments. For sampling linear bivariate causal statements (for the results in Figures 2 and 3), we implement the following procedure:

1. Draw random causal coefficients for $n + m$ variables from a standard normal distribution and error variances for each variable from an exponential distribution with variance 1;

2. Set each causal coefficient to 0 independently with probability $1 - p$;

3. Given the remaining causal coefficients, calculate the covariance matrix of all $n + m$ variables, assuming independence between the error terms (resulting in a linear SEM without confounding);

4. Rescale error variances and causal coefficients so that each variable has unit variance;

5. Select $m$ variables uniformly at random to be the hidden variables and marginalize the covariance matrix and the causal coefficients (using Lemma 2.1) to the remaining $n$ variables.

6. Obtain the correct linear bivariate causal statements from the equation $A = (I - \Gamma)^{-1}$ (see Lemma 2.3).

7. Perturb each bivariate causal statement by adding centered Gaussian noise with variance $\sigma$.

Our procedure for graphical bivariate causal statements for the results in Figure 5 is the following:

1. Draw a uniformly random permutation on $\{1, \ldots, n + m\}$ to get a causal ordering;

2. Set each directed edge that is consistent with the ordering independently with probability $p$ to get a causal DAG;

3. Choose the set of $m$ hidden variables uniformly at random and marginalize the graph to the remaining $n$ observed variables according to Definition 3.2;

4. Obtain the correct graphical bivariate causal statements by further marginalizing the causal graph to each possible pair of variables.

5. Add errors by choosing a possible directed or bidirected edge between all variables uniformly at random and adding it or deleting it from the statement graph depending on whether it was absent or present.

### D.2. Data and Large Language Models

The data for the 7 country-level indicators that we consider is obtained from the gapminder dataset Gapminder Foundation (2026). We use the following detailed variable descriptions for our experiments:

- population density: Average number of people per square kilometer of land in the given country,

- literacy rate: adult literacy rate is the percentage of people ages 15 and above who can, with understanding, read and write a short, simple statement on their everyday life,

- daily income: mean daily household per capita income or consumption expenditure in constant international dollars,

- sanitation access: percentage of people using at least basic sanitation services (improved sanitation facilities not shared with other households),

- smoking: percentage of people over age 15 that smoke,

- happiness score: national average response to a happiness survey, with scores ranging from 0 (worst) to 100 (best),

- life expectancy: average life expectancy at birth in years.

*Table 1.* Correlation matrix for the gapminder dataset.

| | Pop. dens. | Literacy | Income | Sanitation | Smoking | Happiness | Life exp. |
|---|---|---|---|---|---|---|---|
| Population density | 1.000 | 0.109 | 0.708 | 0.104 | -0.018 | 0.078 | 0.128 |
| Literacy rate | 0.109 | 1.000 | 0.373 | 0.798 | 0.109 | 0.526 | 0.716 |
| Daily income | 0.708 | 0.373 | 1.000 | 0.381 | 0.019 | 0.745 | 0.424 |
| Sanitation access | 0.104 | 0.798 | 0.381 | 1.000 | 0.190 | 0.656 | 0.817 |
| Smoking | -0.018 | 0.109 | 0.019 | 0.190 | 1.000 | 0.103 | 0.096 |
| Happiness score | 0.078 | 0.526 | 0.745 | 0.656 | 0.103 | 1.000 | 0.737 |
| Life expectancy | 0.128 | 0.716 | 0.424 | 0.817 | 0.096 | 0.737 | 1.000 |

We show the empirical correlation matrix of these variables in Table 1. We selected these variables as they exhibit a variety of pairwise correlation strengths and causal structure: most notably, population density was chosen as a variable that likely only has weak direct causal relations with many of the other variables, but there is still strong confounding with daily income. On the other hand, there likely are strong causal pathways such as literacy rate → daily income → happiness score. While we accessed all data through https://www.gapminder.org/data/, the data for some of the variables is originally from different sources, shown in the table below. All data is made available through the CC-BY license.

| Data | Source |
|---|---|
| Population Density | United Nations (2024) |
| Literacy Rate | UNESCO Institute for Statistics (2024) |
| Access to sanitation | World Bank Group (2025) |
| Smoking | World Health Organization (2026) |

Table 2 shows the large language models that we query. All models were accessed through Amazon Bedrock and we used the following parameters (except that the Claude Opus models do not allow for specifying top P):

- Maximum number of output tokens: 2048

- temperature: 0.6

- top P: 0.7.

### D.3. Prompts

**Prompt for linear causal statements:** We use the following conversation to elicit linear bivariate statements from the LLMs that we test.

SYSTEM: You are a causality expert, tasked to estimate standardized TOTAL causal effects between country development indicators. Return your answer in HTML format:

*Table 2.* Large language models used in our experiments.

| Model name | Model ID | Provider |
|---|---|---|
| Claude Opus 4.5 | claude-opus-4-5-20251101-v1:0 | Anthropic |
| gpt-oss-120b | gpt-oss-120b-1:0 | OpenAI |
| gpt-oss-20b | gpt-oss-20b-1:0 | OpenAI |
| Gemma 3 27B IT | gemma-3-27b-it | Google |
| Gemma 3 4B IT | gemma-3-4b-it | Google |
| Kimi K2 Thinking | kimi-k2-thinking | Moonshot AI |
| Mistral Large 3 | mistral-large-3-675b-instruct | Mistral AI |
| Magistral Small 2509 | magistral-small-2509 | Mistral AI |
| Qwen3 235B A22B 2507 | qwen3-235b-a22b-2507-v1:0 | Qwen |
| Qwen3 Next 80B A3B | qwen3-next-80b-a3b | Qwen |

<answer>CAUSAL_COEFFICIENT: <number ></answer>

For example: <answer>CAUSAL_COEFFICIENT: 0.35</answer> or <answer>CAUSAL_COEFFICIENT: -0.62</answer>

The causal coefficient quantifies the expected change of the effect variable in standard deviations, given an intervention that changes the cause variable by 1 standard deviation. It includes the effect of all direct causal pathways from the cause to the effect variable. Do not assume away confounding; use realistic domain knowledge. No other text.

USER: I have observational data on 7 country-level development indicators.

Correlation matrix: {corr}

Variable descriptions: {var_names, var_desc}

Before we begin estimating causal coefficients, please determine a plausible causal ordering of these 7 variables (from root causes to downstream effects). Return the ordering as a comma-separated list of variable names inside <ordering> tags. Use the exact variable names shown above.

For example: <ordering>var_a, var_b, var_c, ...</ordering>

ASSISTANT: <ordering>population_density, literacy_rate, daily_income, sanitation_access, smoking, happiness_score, life_expectancy </ordering>

USER: I will now ask you to estimate the total linear causal coefficient for several pairs of variables, following the ordering you provided. For each question, please provide your answer in the format: <answer>CAUSAL_COEFFICIENT: <number></answer>

USER: Estimate the total linear causal coefficient for the causal effect of population_density on literacy_rate.

ASSISTANT: <answer>CAUSAL_COEFFICIENT: 0.1 </answer>

USER: Estimate the total linear causal coefficient for the causal effect of population density on daily income.

ASSISTANT: <answer>CAUSAL_COEFFICIENT: 0.2 </answer>

USER: . . .

Here, the assistant answers are only examples. Moreover, {corr} is a placeholder for the correlation matrix in Table 1 that we provide as a string, {var_names} is a placeholder for the variable names and {var_desc} is a placeholder for the variable descriptions listed in Section D.2. If all assistant answers are in the correct format, then we simply keep asking for the causal effects for all pairs of the seven variables in our dataset, according to the provided causal ordering. If a model answer is not in the correct format, then we send one of the following messages:

USER: Please provide the causal ordering as a comma-separated list of the exact variable names inside <ordering> tags. The variable names are: {var_names}

USER: Please provide your answer in the correct format: <answer>CAUSAL_COEFFICIENT: <number></answer>

These messages are sent up to 5 times in a row until an answer in the correct format is obtained. In our experiments, this was sufficient to obtain answers in the correct format in most runs.

**Prompt for graphical causal statements:** To obtain lists of graphical bivariate causal statements from LLMs, we use the following template for conversation:

SYSTEM: You are a causality expert analyzing relationships between country-level development indicators.

For each pair of variables, you will assess:
1. Whether there is a total causal effect between them, and in which direction
2. Whether there is confounding (correlation not explained by the causal effect between the pair)

Guidelines:
- Answer YES for causal effect if you expect that an intervention on the cause variable would significantly change the effect variable
- Be conservative about confounding - only answer YES for confounding if MOST of the correlation cannot be explained by the causal effect
- Use realistic domain knowledge about socioeconomic factors

Return your answer in the following HTML format:

<causal_effect>
<exists>YES or NO</exists>
<direction>A_TO_B or B_TO_A or NONE</direction>
</causal_effect>
<confounding>
<exists>YES or NO</exists>
</confounding>

Where A_TO_B means the first variable causes the second, and B_TO_A means the second causes the first.

USER: I have observational data on 7 country-level development indicators.

Correlation matrix: {corr}

Variable descriptions: {var_names, var_desc}

I will now ask you about each pair of variables. For each pair:
1. Assess if there is a causal effect
2. Assess if there is strong confounding

USER: Consider the pair: "Population density" (A) and "Literacy rate" (B).

1. Is there a causal effect between them? If yes, in which direction?
2. Is there confounding (significant correlation not explained by the causal effect between these two)?

Provide your assessment in the required format.

ASSISTANT: <causal_effect>
<exists>NO</exists>
<direction>NONE</direction>
</causal_effect>
<confounding>
<exists>NO</exists>
</confounding>

USER: Consider the pair: "Daily income" (A) and "Population density" (B).

1. Is there a causal effect between them? If yes, in which direction?
2. Is there confounding (significant correlation not explained by the causal effect between these two)?

Provide your assessment in the required format.

ASSISTANT: <causal_effect>
<exists>YES </exists>
<direction>B_TO_A</direction>
</causal_effect>
<confounding>
<exists>YES </exists>
</confounding>

USER: ...

If all assistant answers are in the correct format, we keep asking for the causal statements on all pairs of the seven variables

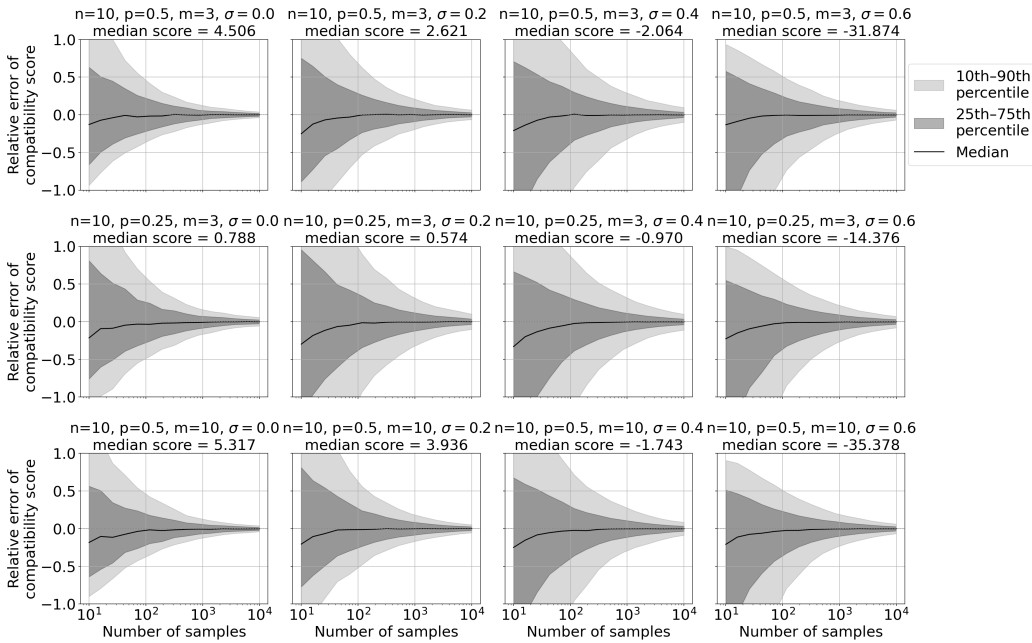

*Figure 8.* Relative error between compatibility scores based on an estimated covariance matrix with finitely many samples and compatibility scores based on the true covariance matrix.

in our dataset. However, we present each pair in a random order, since our graphical incompatibility score also tests the extent to which the causal statements are acyclic. If a model answer is not in the correct format, then we send the following message:

USER: Please provide your answer in the correct format: <causal_effect>
 <exists>YES or NO</exists>
 <direction>A_TO_B or B_TO_A or NONE</direction>
 </causal_effect>
 <confounding>
 <exists>YES or NO</exists>
 </confounding>

Again, sending this message up to 5 times was sufficient to obtain valid model answers.

### D.4. Additional Experiments

First, we present additional plots for the difference between compatibility scores based on empirical and true covariance matrices over different combinations of parameters, see Figures 8 and 9. The plots are obtained as follows:

1. sample 50 different linear Gaussian SEMs (see Appendix D.1),

2. for each model, sample 20 different random perturbations with variance $\sigma$ for the bivariate causal statements,

3. for each model-statement combination, draw iid samples $X \sim \mathcal{N}(0, \Sigma_{\mathbf{X}})$ 10 different times and compute compatibility scores.

As the random error in the statements grows, the absolute value of the compatibility scores becomes larger, and more samples might be required to accurately approximate the true compatibility score. However, since only the sign matters for falsification, an accurate approximation is not always needed, and in fact, the signs converge very quickly across all tested combinations of model parameters.

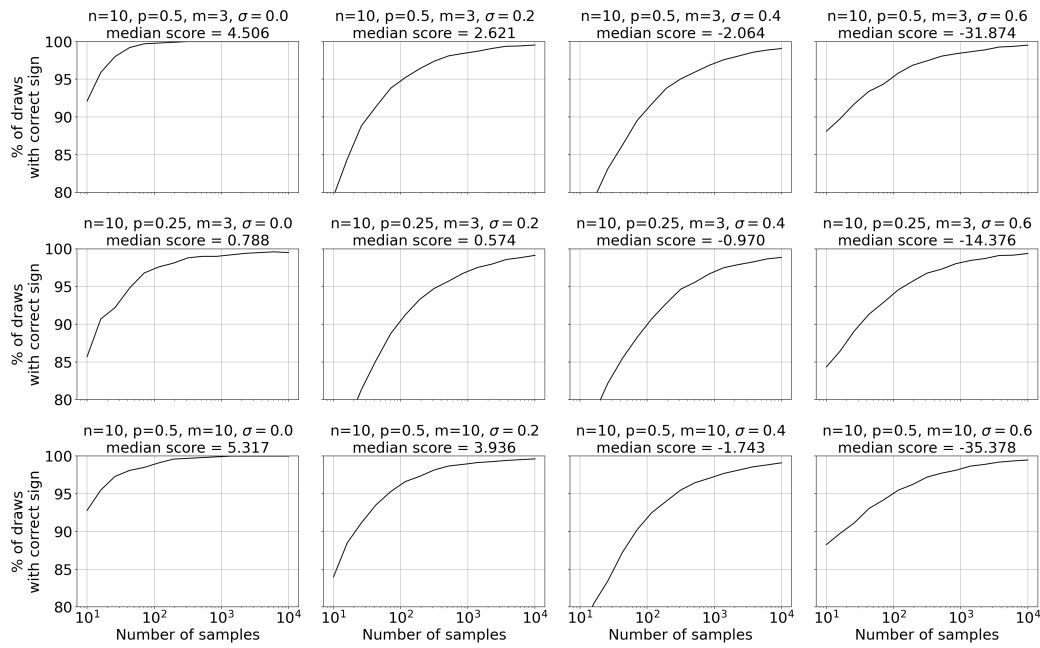

*Figure 9.* Percentage of draws where the compatibility score based on an empirical covariance matrix and the compatibility score based on the true covariance matrix have the same sign.

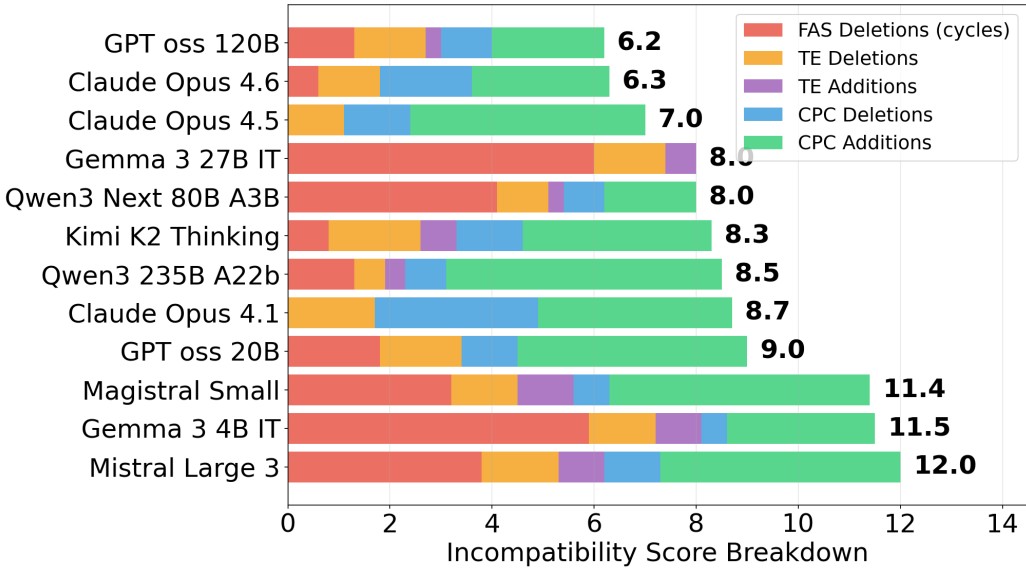

*Figure 10.* Components of the incompatibility score. Here, FAS deletions is the number of edges that our algorithm deleted to make the statement graph acyclic. TE deletions and TE additions refer to the edge modifications to make the statement graph transitively closed, and CPC deletions and CPC additions refer to the edge modifications to make the statement graph satisfy the third property of Lemma 3.5.

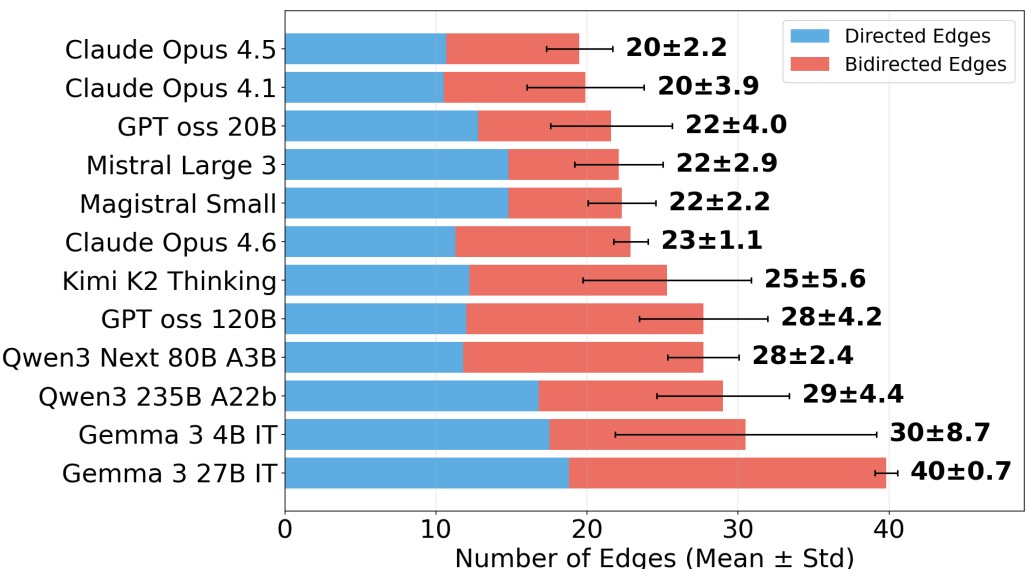

*Figure 11.* Average number of edges of the statement graphs from each model.

Next, accompanying our results in Section 3.4, we provide a breakdown of the average incompatibility scores into the different components calculated by our heuristic algorithm (see Equation (5)) in Figure 10. Notably, Gemma 3 27B IT is the only model that does not have inconsistencies with respect to its bidirected edges. However, a closer look (Figure 11) reveals that it is also the model that reports complete or almost complete graphs in all runs. Indeed, the third property of Lemma 3.5 is trivially satisfied when there is a bidirected edge between every single pair of variables. We tried to avoid this situation by formulating the prompt for graphical causal statements in a way that encourages models to be conservative with placing bidirected edges. However, there is still a large variance in the density of the statement graphs for different models. When focusing on the subset of statement graphs with density at most $2/3$, the lowest achieved incompatibility score is $4$ for a statement graph given by Claude Opus 4.6. For reference, we show this graph in Figure 12, but we remark that a low incompatibility score does not imply correctness of the graph.

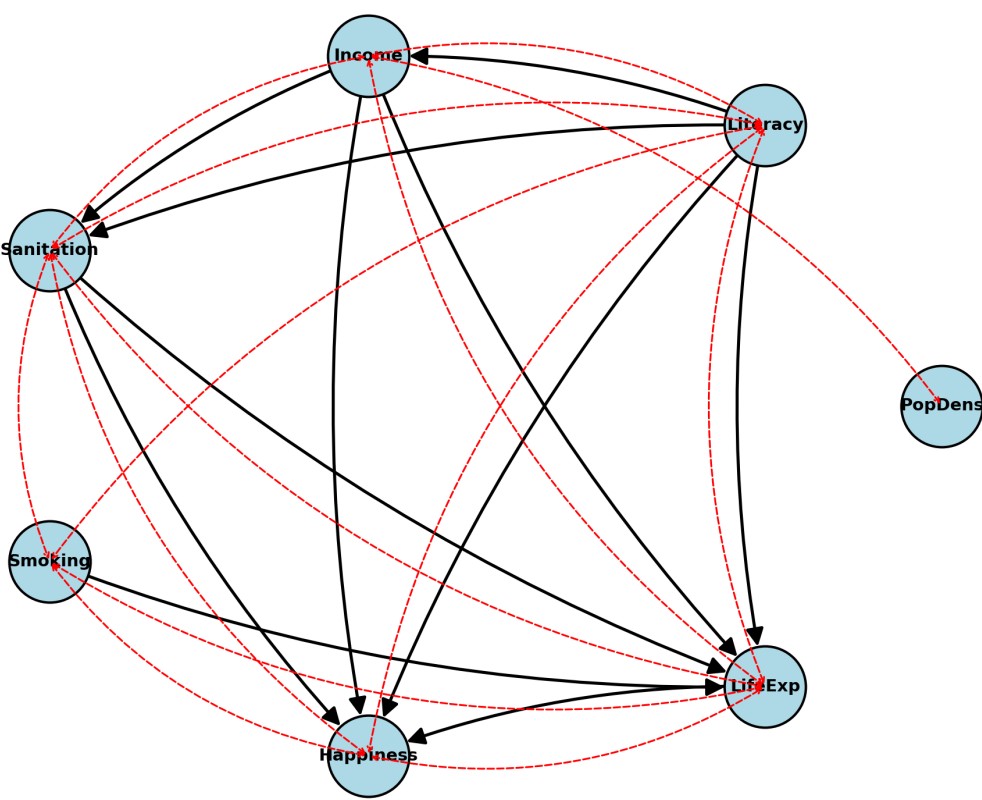

*Figure 12.* Statement graph obtained from Claude Opus 4.6 with incompatibility score 4, 11 directed edges and 14 bidirected edges.

