# OpenReview forum: "Evaluating Bivariate Causal Statements Based on Mutual Compatibility"
_ICML.cc/2026/Conference — ICML 2026 regular_

### Official Review · Reviewer_wWU5 · 2026-02-19

**Soundness:** 3
**Presentation:** 3
**Significance:** 2
**Originality:** 3
**Overall Recommendation:** 5
**Confidence:** 3

**Summary:**

This paper proposes a compatibility score to evaluate causal statements in the setting of linear causal models. They further propose an incompatibility score for cases where quantitative causal statements are unavailable. They validate the effectiveness of their scores on synthetic and LLM-generated datasets.

**Compliance With Llm Reviewing Policy:**

Affirmed.

**Final Justification:**

The authors have addressed my concerns.

**Key Questions For Authors:**

1. I wonder whether it is possible to identify which causal statements contribute to a poor compatibility score.
2. Could the authors provide some discussion on the relationship between this work and causal discovery, where researchers try to estimate causal relationships or SCMs directly from observational data?
3. The font size in the figures should be increased.

**Limitations:**

Yes

**Strengths And Weaknesses:**

**Strength**
1. The paper is well-written, and the problem is clearly defined.
2. The proposed framework is well-motivated. It is important to investigate how to assess the correctness of causal statements when limited ground-truth knowledge is available.
3. The theoretical results are solid and presented well.

**Weakness**
1. Several factors limit the practical utility of the proposed method

* The linear SCM assumption. Real-world data often violates linearity. How useful would this score be when the SCM is non-linear?

* The known causal ordering assumption. How sensitive are the results to the choice of ordering? Could errors in ordering dominate error over other factors?

* High compatibility does not necessarily imply the correctness of the causal statements. This could lead to confusion in practice.

2. In the empirical study, the random baseline outperforms several LLM-based methods. This could make it hard to trust the score in practice. I wonder if the authors could provide any guidance for practitioners.

Overall, I think this paper presents some interesting results on an important problem. However, how useful this metric is in practice remains questionable.

---

> ### Author Rebuttal · Authors · 2026-03-31
>
> Thanks a lot for your review!
> 1. We agree that linear causal ground truth is rare in real-world settings. However, linear causal statements are still very common in causal inference. Hence, we believe that a method for scoring such statements has high practical applicability. Note that our method can provide correct falsification for linear statements without assuming linearity on the causal ground truth. Indeed, if a set of linear statements receives a negative compatibility score, it is either due to a violation of Assumption 2.5 or due to a violation of linearity - in both cases the statements are likely incorrect.
> 2. We would like to emphasize that the true causal ordering does not need to be known for our method to be applicable. We only require a set of linear bivariate causal statements to be consistent with an arbitrary ordering. Moreover, it is possible for statements with a wrong ordering to achieve low compatibility scores. Hence, we do not expect the choice of the ordering to be the dominant factor for the compatibility score. To test this, we repeated our LLM experiments from section 2 with the following twist: Instead of forcing each LLM to produce causal statements with respect to the same fixed ordering, the LLM has to choose the causal ordering itself in the beginning of each run. The results are indeed similar again to our original results, see https://anonymous.4open.science/r/bivariate-causality-4AC0/compatibility_scores_v2.png,  confirming that compatibility scores are not too sensitive with respect to the choice of ordering.
> 3. Our guidance for practitioners using our score can be summarized as follows: A negative score is evidence against the correctness of the given set of statements. A positive score is inconclusive regarding the correctness of the given statements.
> 4. Detecting those causal statements that contributed most to a negative score could be indeed possible: in a first synthetic experiment, we generated random causal models ($n‎ = 10$, $m=3$, $p=0.5$) and added random error to the true bivariate causal statements (as in the experiments shown in Figure 3 of the paper). Then, for each statement separately, we find the argmax that optimizes the compatibility score, keeping all other statements fixed. We greedily pick the statement that improves the compatibility score the most, change it and iterate until the compatibility score is positive. The results show that the statements that got picked in this way did indeed have much higher true error than the other statements, see https://anonymous.4open.science/r/bivariate-causality-4AC0/greedy_recovery.png. This means that our methods could provide a basis for identifying wrong causal statements in a list of statements, which would be an interesting subject for further research.
> 5. We believe that our methods are most effective for evaluating lists of bivariate causal statements that could be obtained from humans, LLMs, or bivariate causal discovery methods (but ideally independently from each other). If one has access to a causal discovery method that can produce causal models on arbitrary subset of variables, our method is still applicable, but it might be more fruitful to evaluate compatibility across larger subsets than just pairs (this is developed in Faller et al. https://proceedings.mlr.press/v238/faller24a.html)
> 6. Thank you for catching the font size being too small in the figures! We will fix this in the camera-ready version.

---

> > ### Author Rebuttal · Reviewer_wWU5 · 2026-04-03
> >
> > Thanks a lot for the clarification and the rebuttal! I don't have any furhter questions and I have adjusted my score.

---

### Official Review · Reviewer_XdAu · 2026-03-02

**Soundness:** 3
**Presentation:** 3
**Significance:** 3
**Originality:** 3
**Overall Recommendation:** 4
**Confidence:** 4

**Summary:**

This paper proposes a framework for evaluating collections of bivariate causal statements when causal ground truth is unavailable. The motivating scenario is one where humans or large language models (LLMs) produce pairwise causal claims (e.g., “X causes Y with coefficient α”), but no experimental validation is possible. The central question is how to assess the reliability of such statements.

The framework is validated via: Synthetic experiments for both linear and graphical settings. Application to LLM-generated causal statements over 7 country-level variables (Gapminder data), showing substantial differences in compatibility scores across models. This paper strives to explore a notable aspect of causal evaluation: internal coherence of causal claims in the absence of ground truth.

**Compliance With Llm Reviewing Policy:**

Affirmed.

**Final Justification:**

The rebuttal addressed my main concerns. I adjust my scores accordingly.

**Key Questions For Authors:**

- How sensitive is the linear compatibility score to finite-sample covariance estimation error?

- Can the confounding postulate be extended to nonlinear SEMs (e.g., additive noise models)? Does uniqueness of induced global models still hold?

- The paper mentions future work on incomplete lists. Is there a principled way to regularize missing edges via compatibility maximization?

- How does compatibility scoring differ from evaluating the likelihood or BIC of the induced SEM?

**Limitations:**

- The linear compatibility guarantee depends on specific generative assumptions (unbiasedness, independence of mechanisms). If real-world processes violate these, compatibility sign may not be informative.

- Linear compatibility assumes a consistent ordering. Real LLM outputs may violate acyclicity.

- It is unclear how the framework scales to dozens or hundreds of variables.

**Strengths And Weaknesses:**

Strengths
1) The paper identifies a practically important but underexplored setting: evaluating causal claims without ground truth, particularly in the context of LLM outputs. The framing is well motivated in the introduction.

2) The observation that any acyclic complete set of linear pairwise total effects induces a unique multivariate SEM is clean and nontrivial. This eliminates naïve compatibility constraints and motivates a deeper notion of plausibility.

3) The idea that marginalization should increase confounding, and that global models requiring additional fine-tuned confounding are implausible, is conceptually appealing. The formalization via Wright’s path decomposition is technically sound.

Weaknesses:
1) The linear compatibility framework: Assumes acyclicity; Requires a known joint covariance matrix; Relies on linear Gaussian SEMs for theoretical justification. This substantially narrows applicability. Real-world causal claims (especially from LLMs) are unlikely to satisfy linearity or correct causal ordering.

2) The squared residual covariance definition is reasonable but not canonical. Alternative confounding measures could yield different behavior. The choice is justified pragmatically but not deeply compared to alternatives.

3) The heuristic for graphical incompatibility is reasonable but lacks approximation guarantees and empirical comparison to optimal small-scale solutions. For dense graphs, overestimation becomes substantial.

---

> ### Author Rebuttal · Authors · 2026-03-31
>
> Thanks a lot for your review!
>
> 1. We agree that linear causal ground truth is rare in real-world settings. However, linear causal statements are still very common in causal inference. Hence, we believe that a method for scoring such statements has high practical applicability. Note that our method can provide correct falsification for linear statements without assuming linearity on the causal ground truth. Indeed, if a set of linear statements receives a negative compatibility score, it is either due to a violation of Assumption 2.5 or due to a violation of linearity - in both cases the statements are likely incorrect.
> 2. It is true that the choice of the confounding score used in our plausibility assumption is non-canonical. We also considered confounding scores based on information-theoretical measures, but these come with the problem that bivariate and multivariate confounding scores often have different scales and are hard to compare. We have simply chosen a measure of plausibility that turned out to be convenient to work with in our evaluations, but we are not dogmatic about this particular choice. We think the main contribution is the idea of quantifying plausibility of the overall model in a simple way after observing that no hard sense of incompatibility exists (unless the causal directions don't match a DAG).
> 3. Our heuristic algorithm comes with two guarantees: First, if the set of statements is graphically compatible, then it outputs incompatibility score 0. Second, it never underestimates the true incompatibility score. This implies that both completely consistent sets of statements and very inconsistent sets of statements are correctly identified from the heuristic score. In the middle regime it is true that our heuristic score can be erroneous. We think that our contribution for graphical incompatibility mainly lies in developing the concepts and providing a first simple algorithm - but the algorithm can certainly be further improved in future work (although an optimal solution is infeasible due to NP-hardness).
> 4. Note that the compatibility score is a quadratic function in the entries of the correlation matrix. As these entries are between 0 and 1, the function is Lipschitz-continuous on the space of correlation matrices. Here, the Lipschitz constant $C$ depends polynomially on the number of variables $n$ and the maximum contribution of any active causal pathway $\max_P |\Gamma^P|$. Now, a standard concentration result gives that $O(\frac{C^2 \log(n)}{\varepsilon^2})$ iid samples are sufficient so that the compatibility score of the empirical covariance matrix differs by at most $\varepsilon$ from the compatibility score of the true covariance matrix. We also validated the robustness of the compatibility score with respect to finite-sample estimation of the covariance matrix in further synthetic experiments, see our answer to reviewer X3ev.
> 5. Uniqueness may not hold anymore for nonlinear SEMs: Consider independent binary variables $X_1, X_2$ and set $X_3 = X_1 \oplus X_2$. Then, the pairwise marginals are the same as for the model that simply consists of three independent binary variables. However, we believe that checking whether our approach can be generalized for specific restricted families of nonlinear models would be an interesting direction for future research (additive noise models are probably not the right family, since they are not closed under marginalization)
> 6. See our answer to reviewer wWu5 for how optimizing individual statements for maximal compatibility could be a promising approach.
> 7. Likelihood is not useful in our setting of scoring a set of causal statements under arbitrary confounding because for any matrix of causal coefficients $\Gamma$ there is a choice of the noise vector $\mathbf{N}$ such that the induced vector of observables $\mathbf{X} = (I-\Gamma)^{-1} \mathbf{N}$ perfectly fits the empirical covariance matrix and hence achieves maximum likelihood (so we already implicitly assume that a set of bivariate causal statements induces the noise vector that maximizes the likelihood of the model). The BIC score is likelihood together with an $\ell_0$-penalty for the model parameters - since most sets of bivariate causal statements won’t induce a parameter that is exactly zero, the BIC score will again be equal for those sets. Of course, one could instead use an $\ell_1$-penalty or similar, but such a score would implicitly assume that SEMs with larger coefficients are less plausible. In our opinion, this is less believable than our definition of plausibility given by Assumption 2.5, which our compatibility score is based on.

---

> > ### Author Rebuttal · Reviewer_XdAu · 2026-04-01
> >
> > The rebuttal responds to my concerns.

---

### Official Review · Reviewer_X3ev · 2026-03-12

**Soundness:** 3
**Presentation:** 2
**Significance:** 2
**Originality:** 2
**Overall Recommendation:** 4
**Confidence:** 3

**Summary:**

The paper focuses on evaluating the quality of bivariate causal statements over a set of variables. Two criteria are proposed. The first criterion works for multivariate linear SEMs, which is based on comparing the amount of confounding remaining in the collection of bivariate causal graphs and the multivariate graph. The second criterion is qualitative and can be applied to more general ADMGs. In particular, for a statement graph $\mathcal G$, it defines the compatibility score to be the minimal Hamming distance between $\mathcal G$ and some $\mathcal G^*$ satisfying graphical compatibility. To solve the optimization problem, the authors provide a heuristic, greedy algorithm. Experiments are conducted to show their ability to falsify causal statements.

**Compliance With Llm Reviewing Policy:**

Affirmed.

**Final Justification:**

The authors have successfully addressed most of my concerns, and therefore, I have raised the score to 4. I did not give a higher score because the theoretical guarantee is limited, since Q3 was only partially addressed.

**Key Questions For Authors:**

Additionally, I am confused about Figure 3. $m$ is the number of hidden variables, and it was set to 10 for a line in the left figure. But there are a total of 10 variables.

**Limitations:**

Yes

**Strengths And Weaknesses:**

**Strengths:**
1. The paper tackles an interesting problem of falsifying causal statements, which might be useful for practitioners.
2. It has some novelty, e.g. generalization of compatibility measures and the optimization algorithm.
3. The description of the methodology is clear and easy to understand.

**Weaknesses:**
1. I found the utility of the proposed criteria sort of vague and needs more clarification. For example, it seems that the ordering of the variables (or a statement graph) is needed to compute the compatibility score. Thus, if I provide a wrong topological ordering, then it is likely that the incompatibility would be high. But unfortunately, even in a linear SEM with a couple of variables, the chance to select the true(allowed) topological ordering can be very small. Moreover, it can not tell us which specific statement is wrong. Therefore, I feel like the criteria are more suitable for evaluating the output of real causal discovery methods, since the true hamming distance to the underlying graph is never known in practice. I highly recommend that the authors conduct experiments in this direction and check whether the output is highly correlated with the true SHD.
2. The above also makes me confused about the experiments on the LLMs. Suppose the topological ordering/statement graph given is wrong, then shouldn't the LLM ideally output high incompatibility? Likewise, low incompatibility is expected if the ordering is correct. However, it seems not clear whether the ordering in the experiments is the ground truth, potentially making the performance comparison in the paper less meaningful. I suggest that the authors consider datasets with known causal tiers, e.g. https://github.com/cmu-phil/example-causal-datasets.
3. The authors wanted to show the soundness of the criterion in Theorem 2.10, by proving that the expected compatibility is positive when the data is generated by a multivariate linear SEM. However, this is not enough. A more reasonable approach is to show that the tail bound, i.e. the probability of having negative compatibility, is negligible. Because the mean is easily affected by extreme values.
4. There are more concerns about the experiments in Figure 3. In practice, the coefficients are learned from empirical data, and therefore, it would be more valuable to plot the percentage against the number of samples available for a causal discovery/regression method to learn the coefficients. This would tell us how many samples are needed for the proposed criterion to produce a meaningful output. In particular, I wonder the percentage of positive compatibility(likewise, the mean/median of the compatibility) when the coefficients are estimated by a common method based on a certain number of samples?

---

> ### Author Rebuttal · Authors · 2026-03-31
>
> Thanks a lot for your review!
>
> 1. We would like to emphasize that knowing the true causal ordering is not needed to apply our methods. Only for quantitative statements, we require consistency with some (arbitrary) ordering. It is not necessarily the case that statements with a wrong ordering have a low compatibility score. This is why we state that high compatibility scores never show correctness, only low compatibility scores can be used as evidence against the correctness of a list of causal statements. Given a negative score, it could actually be possible to identify those statements that induce the most error, see our answer to reviewer wWu5. We appreciate the suggestion of comparing our graphical incompatibility score with SHD but it does not quite work in our setting, since the SHD requires a (multivariate) estimate graph, but a list of bivariate graphical statements does not necessarily correspond to a unique such graph. Instead, we compare the incompatibility score to the number of errors in the bivariate statements (which can be thought of as the SHD equivalent in our setting) in Figure 4.
>
> 2. We agree with you that it is unclear how forcing a fixed causal ordering affects the results of our LLM experiments (this is for quantitative statements, note that we do not force any ordering in the experiments with qualitative statements). That’s why we repeated our experiments with quantitative LLM statements adding a prompt in the beginning that asks each LLM to choose the causal ordering itself. Still, the results are quite similar to our original results, see https://anonymous.4open.science/r/bivariate-causality-4AC0/compatibility_scores_v2.png, confirming that the ordering is probably not the dominating factor for the compatibility score of a list of bivariate statements.
>
> 3. It’s true that a good tail bound would further improve Theorem 2.10. We found that proving a strong concentration result likely requires more assumptions on the distribution over true causal models. While our result is weaker, we think that the strength of our Theorem lies in the fact that we do not have to impose a strict view on how true causal models typically look like, but instead have a result for a very broad class of distributions. We still demonstrate empirically that concentration holds, I.e. the compatibility score across a variety of parameter settings for random causal models is positive with high probability (these are the data points in Figure 3 where the error of the statements is 0).
>
> 4. Thank you for the suggestion of testing our method in the finite-sample setting. We conducted further synthetic experiments, where a random causal model is drawn, a list of samples is generated for the model and we report the relative difference of the compatibility scores of the true causal statements using the true covariance matrix vs using the empirical covariance matrix, see  https://anonymous.4open.science/r/bivariate-causality-4AC0/sensitivity_rel_error.png. We also report the difference of scores for causal statements that are obtained by regression based on either the true covariance or the estimated covariance, see https://anonymous.4open.science/r/bivariate-causality-4AC0/sensitivity_rel_error_estimated.png. The results show that the scores converge relatively quickly, with 100 samples being sufficient to keep the relative difference below 20% with probability at least 90% in most settings. However, sparse graphs require more samples for a similar accuracy (but also see our answer to reviewer XdAu for a theoretical discussion).
>
> 5. In Figure 3, $n$ is the number of observed and $m$ is the number of hidden variables, so for $n=10$, $m=10$, there are 20 variables in total. Thank you for pointing out the potential confusion.

---

> > ### Author Rebuttal · Reviewer_X3ev · 2026-04-03
> >
> > I appreciate the authors for the detailed rebuttal. Q1,Q2 are fully resolved, and Q3 is partially addressed, as the theory is not provided.
> >
> > For Q4, I appreciate the experiments. I wonder if the authors could also provide a similar convergence illustration of a wrong statement that can be identified by a negative compatibility score (on the population level). I will increase my score to 4 if the result looks similar to the existing ones.

---

> > > ### Author Response · Authors · 2026-04-04
> > >
> > > Thank you very much for your answer!
> > >
> > > We have expanded our experiments as follows:
> > > - draw multiple random ground truth models
> > > - draw multiple iid samples for each model
> > > - apply random perturbations to the true causal statements
> > > - compare the compatibility scores for the erroneous causal statements using a) the true covariance matrix and b) the estimated covariance matrix based on the samples
> > >
> > > We report the relative difference of the compatibility scores here: https://anonymous.4open.science/r/bivariate-causality-4AC0/sensitivity_rel_error_grid.png
> > > And the percentage of draws where both scores have the same sign (because the sign is our plausibility criterion): https://anonymous.4open.science/r/bivariate-causality-4AC0/sensitivity_same_sign_grid.png
> > >
> > > We find that in most test settings up to 1000 samples can be necessary to approximate the true score with error less than 25%. However, for larger perturbations, the scores can be very negative (<-1000), so a precise approximation is not necessary to draw the correct conclusion for falsification. When we focus on the sign, we find that across all settings, roughly 100 samples are already sufficient for the signs of the scores to agree in >90% of all cases.
> > >
> > > We hope that this sufficiently answers your remaining concerns! In case there is any unclarity about the experiment, the source code can be found here https://anonymous.4open.science/r/bivariate-causality-4AC0/sensitivity_experiment.py

---

### Official Review · Reviewer_8ukL · 2026-03-13

**Soundness:** 3
**Presentation:** 3
**Significance:** 3
**Originality:** 3
**Overall Recommendation:** 5
**Confidence:** 3

**Summary:**

The authors present (in)compatibility scores for measuring consistency in a collection of bivariate causal statements. Statements encoding structural and parametric properties of the causal model might be globally inconsistent in terms of confounding. Derivations, theoretical properties and en empirical evaluation with LLMs is presented.

**Compliance With Llm Reviewing Policy:**

Affirmed.

**Key Questions For Authors:**

N/A (Thanks.)

**Limitations:**

Yes

**Strengths And Weaknesses:**

(+) Important avenue in causality research: effectively testing for plausibility in the light of confounding can serve for better causal discovery in the future.

(+) Complete coverage in the paper (motivation, derivation, properties, evidence)

(+) The authors proof a correspondence between the multivariate SEM and collection of bivariate SEMs perspectives

(+) The authors define a compatibility score on a list of bivariate causal statements based off well-known confounding measure decompositions

(-) The authors assume a linear SEM whose causal effects can be summarized within a real-valued matrix, however, any work will be based off assumptions (even if unrealistic at times) and no single paper solves every problem in the world, which is why this does not alter my final opinion

(+) Theorem 2.10 is a strong result suggesting for arbitrary dimensions (note that bivariate is excluded since we are considering multivariate to bivariate relations) that under the assumptions in 2.9 we can expect non-zero compatibility

(+) I did not run the code myself but the authors even provide all their code in a repo that seems alright from swift skimming.
Seeing a separation in LLM performance inline with the characterization of the score is nice, although short!

(-) Rather Assumption 2.5 is a tricky one, but with what we have seen in the literature thus far, rejecting based on this IMHO is too harsh. However, of course we can have a discussion on this matter

---

> ### Author Rebuttal · Authors · 2026-03-31
>
> Thanks a lot for your review! While linearity is indeed often violated in real-world causal systems, linear causal statements are still very common. This is why we believe that a method for scoring linear causal statements is very useful in practice, and we totally agree with your view that some type of assumption is usually always necessary to solve difficult problems such as evaluating causal statements without ground truth.

---

### Decision · Program_Chairs · 2026-04-30

**Decision:**

Accept (regular)

**Comment:**

The reviewers seem to agree that the paper proposes an interesting and novel method for evaluating the consistency of a set of bivariate causal statements with applications to LLMs, which is a timely topic. The paper is well-written and well-presented, although the reviewers noted some of the assumptions might limit its practical applicability, e.g. the assumption of linear causal statements. Overall, this is a very solid paper and I'm sure addressing the reviewers' suggestions will further improve its final version.